# Sensory conflict disrupts circadian rhythms in the sea anemone *Nematostella vectensis*

**Cory A Berger**[1,2]*, **Ann M Tarrant**[1]*

[1]Biology Department, Woods Hole Oceanographic Institution, Woods Hole, United States; [2]MIT-WHOI Joint Program in Oceanography/Applied Ocean Science & Engineering, Woods Hole, United States

**Abstract** Circadian clocks infer time of day by integrating information from cyclic environmental factors called zeitgebers, including light and temperature. Single zeitgebers entrain circadian rhythms, but few studies have addressed how multiple, simultaneous zeitgeber cycles interact to affect clock behavior. Misalignment between zeitgebers ('sensory conflict') can disrupt circadian rhythms, or alternatively clocks may privilege information from one zeitgeber over another. Here, we show that temperature cycles modulate circadian locomotor rhythms in *Nematostella vectensis*, a model system for cnidarian circadian biology. We conduct behavioral experiments across a comprehensive range of light and temperature cycles and find that *Nematostella*'s circadian behavior is disrupted by chronic misalignment between light and temperature, which involves disruption of the endogenous clock itself rather than a simple masking effect. Sensory conflict also disrupts the rhythmic transcriptome, with numerous genes losing rhythmic expression. However, many metabolic genes remained rhythmic and in-phase with temperature, and other genes even gained rhythmicity, implying that some rhythmic metabolic processes persist even when behavior is disrupted. Our results show that a cnidarian clock relies on information from light and temperature, rather than prioritizing one signal over the other. Although we identify limits to the clock's ability to integrate conflicting sensory information, there is also a surprising robustness of behavioral and transcriptional rhythmicity.

**\*For correspondence:**
cberger@whoi.edu (CAB);
atarrant@whoi.edu (AMT)

**Competing interest:** The authors declare that no competing interests exist.

## Editor's evaluation

Understanding the integration and contribution of different combinations of environmental cues to the synchronization of the daily oscillator is important, because it provides insight into how organisms might be able to distinguish (and weight) between irregular (or in the tidal zone highly complex) versus regular individual daily changes of light and temperature. The study, which is thoroughly conducted and provides an impressive amount of experimental and analytical work, dissects the effects of sensory conflict on behavior and gene expression rhythms.

## Introduction

Nearly all organisms use circadian clocks to synchronize their behavior, physiology, and metabolism to ~24 hr environmental cycles. Factors that synchronize circadian clocks are called zeitgebers, and can include light, temperature, and food availability (*Aschoff, 1960*). In nature, clocks must integrate information from many zeitgebers simultaneously to infer coherent clock outputs, although most of our knowledge of circadian clocks comes from single-zeitgeber experiments. During 'sensory conflict', when zeitgebers give conflicting information to the clock, circadian rhythms can either be disrupted

**eLife digest** Almost all living things exhibit circadian rhythms – internally driven biological processes – which regulate important bodily functions, including sleep and wake cycles, over a roughly 24-hour period. Circadian clocks govern these rhythms by receiving information from the environment that allows them to tell what time of day it is. Two of the most important environmental signals, known as 'zeitgebers' – meaning 'time giver' – are light and temperature.

In nature, circadian clocks must integrate information from multiple zeitgebers simultaneously. Typically, over a 24-hour period, temperature increases and decreases with the light cycle, getting warmer during the day and colder at night. However, artificial light pollution and circadian disruption – such as shift work – can impact the natural relationship between light and temperature. This 'sensory conflict', where two zeitgebers provide conflicting information about the time of day, can impact ecosystems such as coral reefs; and is also linked to poor health in humans. How circadian clocks behave in complex multi-zeitgeber environments and specifically, whether they prioritize one zeitgeber over another is not fully understood.

To investigate how cnidarians – a group of marine animals including corals and jellyfish – respond to sensory conflict, Berger and Tarrant varied the relationship between light and temperature cycles using the sea anemone *Nematostella vectensis* as a model system. *Nematostella* is a nocturnal cnidarian, meaning it moves most at night. First, Berger and Tarrant kept *Nematostella* in dark conditions with 24-hour temperature cycles – starting cold, increasing to a peak in the middle of the day before decreasing towards the end of the day. Monitoring *Nematostella* movement revealed that they moved most during the cold phase, showing that temperature cycles alone can maintain rhythmic behavior. Similarly, when temperature and light cycles were aligned such that both rose and fell together, nocturnal behavior was preserved. However, when large misalignments between light and temperature cycles were introduced – such that temperature decreased during light periods and increased in the dark – nocturnal behavior was almost completely lost. This suggests that both light and temperature interact to produce complex patterns of circadian behavior, with neither signal being prioritized over the other.

Additionally, Berger and Tarrant investigated how sensory conflict impacts the activity of *Nematostella* genes. While many genes remained rhythmic, suggesting some gene expression persists when behavior is disturbed, others that were rhythmic became arrhythmic. In contrast, a selection of genes that do not normally display rhythmic behavior gained rhythmic expression. Genes related to protein metabolism and other energy-intensive processes were particularly disrupted.

In an increasingly 24/7 society, it is important to understand how complex multi-sensory environments impact circadian rhythms and as a result, health and fitness. The findings show that certain light and temperature regimes severely disrupt *Nematostella* behavior and could be useful in predicting how other organisms might respond to disruptions such as light pollution. In the future, such information could be used to design optimal light regimes for ecosystems in which the relationship between light cycles and other environmental signals is disrupted by human behavior.

(*Harper et al., 2016*) or preferentially follow a single zeitgeber (*Harper et al., 2017*). By defining conditions under which normal clock output is possible or disrupted, studies of sensory conflict help us understand how environmental information is transmitted into rhythmic behavior, which is a central goal of circadian biology (reviewed for insects in *Rivas et al., 2015*; *Somers et al., 2018*). The behavior of clocks under conflicting zeitgeber regimes is particularly relevant to human society, where factors such as shift work and artificial light regimes are connected to a wide range of metabolic and neurological diseases (*Ehlers et al., 1988*; *Roenneberg and Merrow, 2016*), and in the context of artificial light pollution impacting natural ecosystems (*Gaston et al., 2013*). However, few studies in any animal have comprehensively characterized clock function in multi-zeitgeber systems, and in general we do not understand how animals behave during sensory conflict.

We use the phrase 'sensory conflict' to refer to a situation in which two environmental signals provide different phase information, as compared to a control in which both signals are aligned in phase. In this narrow sense, there is a small literature of sensory conflict experiments that have examined conflicting light and temperature cycles in insects (*Miyasako et al., 2007*; *Currie et al., 2009*;

*Watari and Tanaka, 2010*; *Yoshii et al., 2010*; *Nikhil et al., 2014*; *Harper et al., 2016*; *Harper et al., 2017*; *Rivas et al., 2018*; *Kaniewska et al., 2020*), vertebrates (*Firth and Kennaway, 1989*; *Valenciano et al., 1997*; *Moyer et al., 1997*; *Firth et al., 1999*), protists (*Bruce, 1960*), and cyanobacteria (*Lin et al., 1999*). There have also been studies of light and time-restricted feeding in vertebrates (*Sánchez-Vázquez et al., 1995*; *Challet et al., 1998*; *Reierth and Stokkan, 1998*; *Lague and Reebs, 2000*). However, many of these studies did not compare more than two possible phase relationships between zeitgebers, precluding characterization of the potentially non-linear relationship between an entrained rhythm and two entraining cues. There is a larger literature concerning interactions of other entraining factors such as tidal and lunar rhythms (e.g. in marine invertebrates, see *Connor and Gracey, 2011*; *Wuitchik et al., 2019*; *Bertolini et al., 2021*), and it is clear that rhythmic cues often interact to produce complex patterns of behavior and gene expression. There is a need for experiments to comprehensively assess phase relationships between zeitgebers to understand the sensitivity of clock outputs to environmental conditions.

Another critical knowledge gap is how sensory conflict affects rhythmic gene expression, which ultimately underlies rhythmic physiology and behavior (*Yeung and Naef, 2018*). Animal clocks function via transcription-translation feedback loops in which transcription factors activate the expression of their own repressors, and also regulate transcription of downstream genes (*Partch et al., 2014*; *Dubowy and Sehgal, 2017*). Although one study measured the expression of individual clock genes during sensory conflict (*Rivas et al., 2018*), the effects on downstream gene expression are unknown. In general, interactions between rhythmic factors can produce wide variation in gene expression and impact a variety of biological processes (*Connor and Gracey, 2011*; *Wuitchik et al., 2019*). Animals exposed to arrhythmic conditions, such as mice in feeding experiments (*Greenwell et al., 2019*) or cavefish over evolutionary time (*Mack et al., 2021*), often simply express fewer rhythmic transcripts. Transcriptome-scale studies are needed to test whether sensory conflict similarly disrupts expression of core clock genes or of downstream clock-controlled genes, resulting in a general loss of rhythmic gene expression.

We address these questions with the model system *Nematostella vectensis*, the starlet sea anemone. Studies of circadian rhythms in Cnidaria, the sister group to Bilateria (*Simion et al., 2017*), are broadly important for understanding how clocks have evolved and diversified in animals. Specifically, cnidarians provide an intriguing system for circadian sensory processing because they lack a central nervous system. In bilaterians, researchers often distinguish central clocks in the brain and peripheral clocks in other tissues, with the central clock acting to synchronize peripheral clocks (*Dibner et al., 2010*). Certain phase relationships between light and temperature cycles disrupt the central clock in *Drosophila* (*Harper et al., 2016*), but not peripheral clocks, which preferentially entrain to light (*Harper et al., 2017*). Thus, sensory conflict might disrupt organism-level behavior because of the differential entrainment of central and peripheral clocks. Cnidarians lack this distinction and therefore provide an important and complementary framework to bilaterian models for studying sensory conflict. *Nematostella* exhibits nocturnal rhythms in locomotion that entrain to artificial light-dark cycles (*Hendricks et al., 2012*; *Oren et al., 2015*) and to natural environmental conditions (*Tarrant et al., 2019*). This species inhabits shallow salt marshes at temperate latitudes and can experience daily temperature swings of more than 20 °C (*Tarrant et al., 2019*; *Sachkova et al., 2020*), suggesting that temperature may be a relevant entraining cue. Although temperature cycles are a nearly-universal zeitgeber (e.g. *Somers et al., 1998*; *Glaser and Stanewsky, 2005*; *Lahiri et al., 2005*; *Yoshida et al., 2009*; *Hart et al., 2021*), no studies have yet investigated thermal entrainment in a non-bilaterian animal.

Here, we show that environmentally realistic temperature cycles drive rhythmic behavior and affect the circadian clock in *Nematostella*, producing activity profiles similar to light-driven rhythms. We then comprehensively test different combinations of light and temperature cycles and show that aligned cycles drive normal rhythmic behavior, while behavior becomes arrhythmic under large degrees of misalignment. In particular, we demonstrate the disruption of free-running rhythms, indicating changes to the endogenous clock itself. Using RNA sequencing, we find that sensory conflict substantially alters diel gene expression patterns. This includes both the loss and gain of rhythmic genes, the disruption of co-expressed gene modules related to cellular metabolism, and an overall weakening of correlations between rhythmic genes. However, much of the transcriptome remained rhythmic and in-phase with the temperature cycle. This study extends our understanding of temperature entrainment and

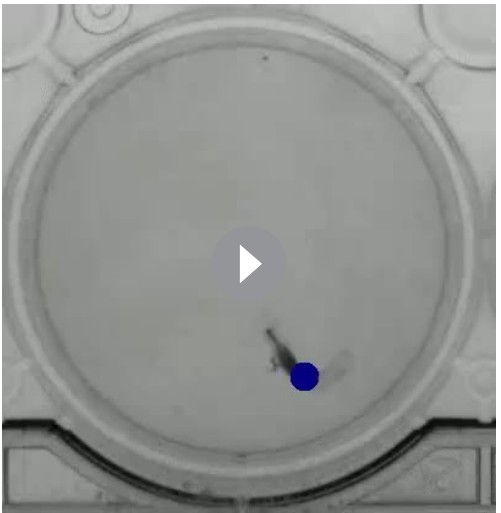

**Video 1.** Example of *Nematostella vectensis* locomotor behavior, with tracked center-point estimated by DeepLabCut. This video is sped up 200 x and covers a period of 100 min. Width of arena is 3.5 cm.
https://elifesciences.org/articles/81084/figures#video1

integration of multiple environmental signals to a non-bilaterian model animal, showing that light and temperature interact to synchronize clock outputs, with neither cue totally dominating the other. Thus, normal organismal rhythms are only possible under certain environmental conditions.

## Results

### Temperature cycles drive circadian rhythms in *Nematostella*

We first showed that gradual, environmentally-relevant temperature cycles drive rhythmic loco-motor behavior and influence circadian rhythms in *Nematostella*. To quantify locomotion, we tracked the center-point of individual anemones using the machine learning software DeepLabCut (*Mathis et al., 2018*) from video files recorded at 2 frames per second over 3 days (*Video 1*). Data were summed into hourly bins and normalized to each animal's maximum hourly movement, producing data consistent with previous studies (*Hendricks et al., 2012*; *Oren et al., 2015*; *Tarrant et al., 2019*). Anemones maintained in constant dark-

ness with a temperature cycle that changed gradually from 14–26 °C (1 degree per hour; *Figure 1a*) exhibited rhythmic diel behavior (n=18, eJTK test, p=3 × $10^{-5}$; LSP permutation test, $p < 0.001$, *Figure 2a*). For temperature cycle experiments, we define ZT0 as the coldest point of the cycle; peak activity occurred at ZT18. Free-running rhythms that persisted during constant temperature at 20 °C were statistically detectable (LSP, $p < 0.001$, n=23) but not visually obvious (*Figure 2a*), suggesting that free-running rhythms, if present, are weak or noisy. We observed similar results for a broader temperature cycle (8–32 °C, *Figure 2a*), and we note that both cycles are within the range of natural daily temperature variation experienced by the source population (*Tarrant et al., 2019*; *Sachkova et al., 2020*).

Although we did not observe strong free-running rhythms in the average behavior profiles, some individuals did have ~24 hr free-running periods (4/23 for the 14–26 °C cycle, 10/46 for the 8–32 °C cycle; eJTK, p<0.001), which we interpret as evidence that temperature cycles can entrain free-running rhythms in *Nematostella*. To further investigate whether temperature is a true zeitgeber, we conducted an additional experiment in which we exposed anemones to a short 12 hr tempera-ture cycle, half the length of a 24 hr circadian period. In published studies in other taxa, organisms entrained to zeitgeber periods half the length of their endogenous period often display two behav-ioral components, one with a 12 hr period corresponding to a directly driven component and one with a circadian period (*Bruce, 1960*; *Merrow et al., 1999*). The circadian component can only be observed if the short period synchronizes an endogenous circadian pacemaker. Therefore, we used periodograms to assess the relative contributions of 12 hr and 24 hr components to the behavior of *Nematostella* in short-period cycles. To validate this approach, we first demonstrated that 24 hr behavior occurs during 6:6 light cycles (LD; *Figure 2—figure supplement 1a*). During 6:6 LD, we found that 8/12 animals had a significant circadian component (LSP, p<0.01) and 6/12 had a circatidal component (4 had both), suggesting that 6:6 LD cycles can entrain circadian behavior (*Figure 2—source data 1*). We noticed that although most animals had circadian behavior, their activity did not line up within 12 hr windows and thus the mean time series did not have a 24 hr component. In other words, because the 6:6 LD cycles are identical, the animals did not all perceive the same 12 hr window as 'day' or 'night'. We therefore aligned all animals by dividing them into two groups based on the 12 hr window in which they had the highest activity. After shifting one group by 12 hr, the new mean time series had a strongly circadian component (LSP, p<0.01). We conclude that 6:6 LD cycles entrain circadian behavior in *Nematostella*, which is expected because light is a strong zeitgeber. We then

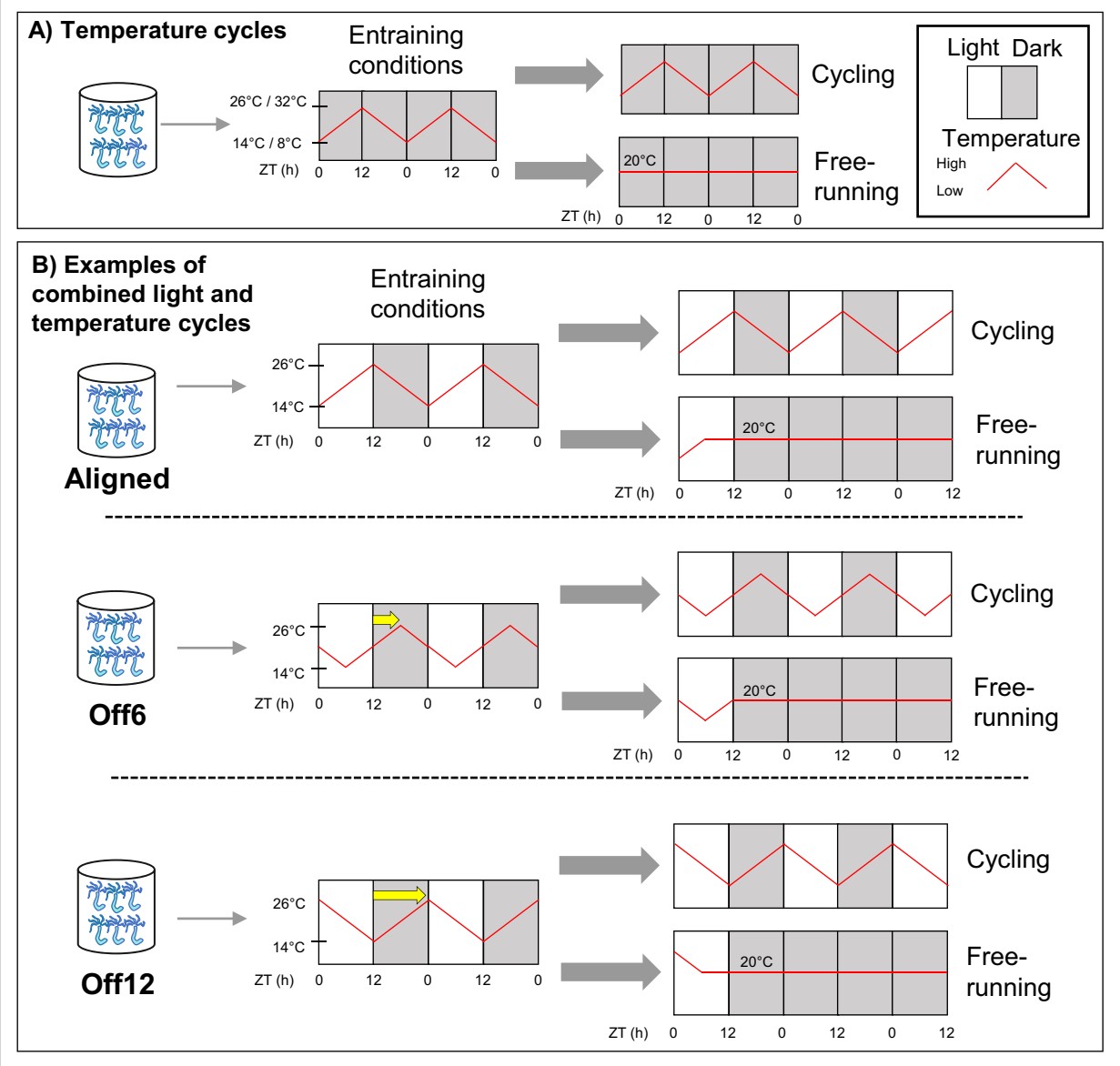

**Figure 1.** Schematic of the experimental design. (**A**) Anemones in constant darkness were entrained to ramped 24 hr temperature cycles that changed gradually from 14–26 °C, or from 8–32 °C. For temperature cycle experiments, ZT0 refers to the coldest time point. Locomotor behavior was recorded during entrainment conditions (n=18 for 14–26 °C, n=17 for 8–32 °C) and for different individuals during free-running conditions (constant darkness and 20 °C; n=23 for 14–26 °C, n=46 for 8–32 °C). We used qRT-PCR to measure the expression of select clock-related genes for the 8–32 °C group (*Figure 2c*), with time points taken every 4 hr for 48 hr during both entrainment and free-running conditions. (**B**) Anemones were entrained to 12:12 light-dark cycles (LD) and simultaneous 14–26 °C temperature cycles. For these experiments, ZT0 always refers to lights-on. In the Aligned (reference) group, the coldest part of the temperature cycle was aligned with lights-on. For six other groups of anemones, the phase of the temperature cycle was delayed relative to the light cycle in 2 hr increments up to 12 hr. For simplicity, we only show the 6 hr (Off6) and 12 hr (Off12) offsets here (yellow arrows indicate phase shifts). For each group, behavior was recorded during entrainment conditions (n=24[a]), and for separate groups of anemones (n=24) during free-running (dark, 20 °C). We conducted RNA sequencing for the Aligned and 12 hr offset groups, with time points taken every 4 hr for 48 hr during entrainment conditions. In both (**A**) and (**B**), anemones were acclimated to experimental conditions for at least 2 weeks before measuring behavior for 3 days, with feeding occurring for each group at the same time of day around ZT8. Animals were not fed during behavior recording. Shaded and unshaded regions represent dark and light periods, respectively, and red lines represent temperature. [a]Except the 12 hr offset, which had n=36.

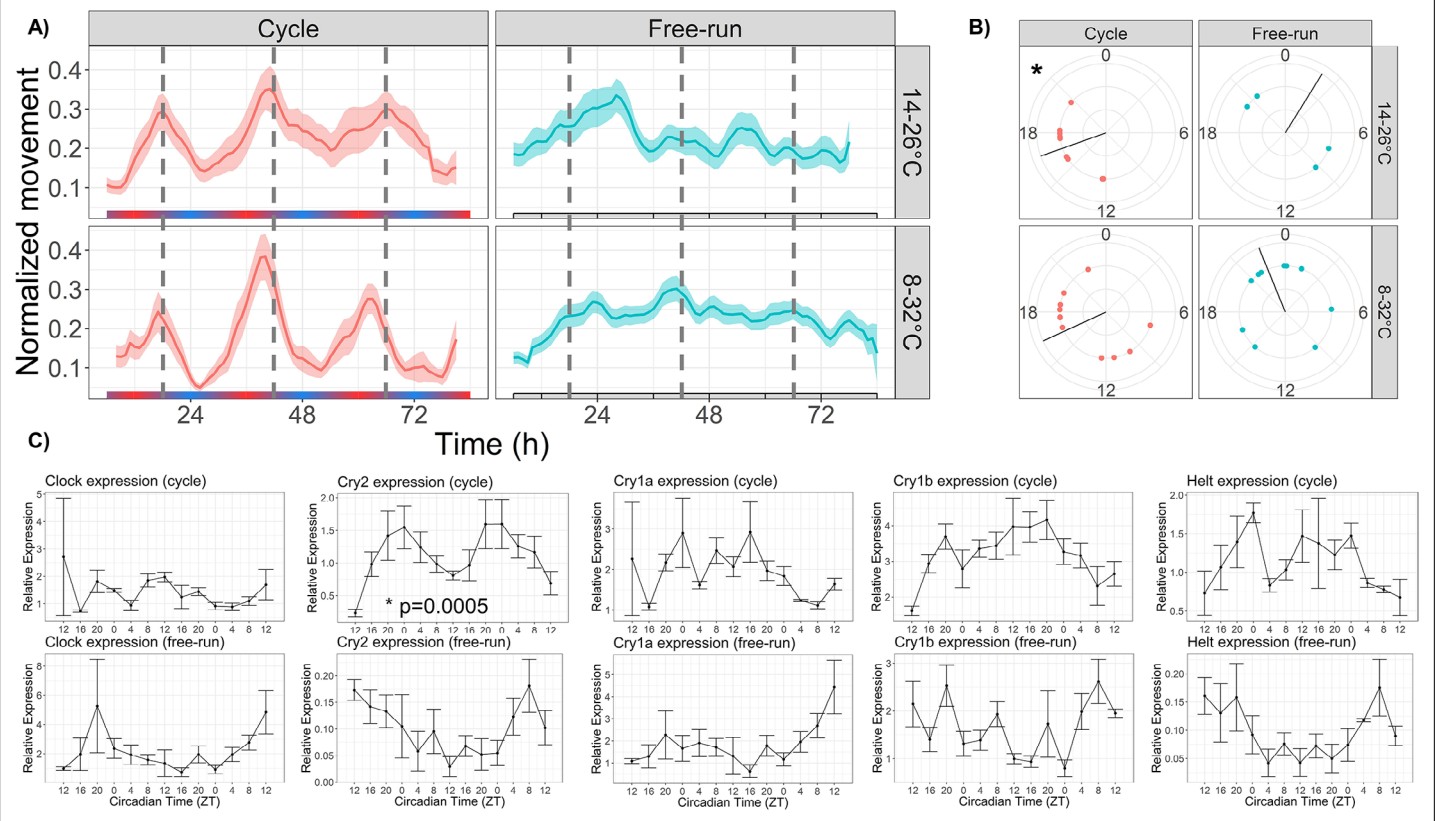

**Figure 2.** Temperature cycles drive rhythmic locomotor behavior and *Cry2* expression, and weakly entrain circadian behavior. (**A**) Mean behavior profiles of anemones in each group over 3 days. Individual locomotor profiles were normalized, averaged, and smoothed (see Materials and methods). Shaded area represents standard error. Left, temperature cycles with scale bar indicating cold (blue) and hot (red) temperatures; n=18 (top) and n=23 (bottom). Right, free-running at 20 °C; n=17 (top) and n=46 (bottom). Dashed lines indicate ZT18, the period of peak activity during entrainment conditions. (**B**) Phases of rhythmic animals calculated by MFourFit. Black line represents circular mean. *:Rayleigh test, p<0.05. (**C**) Expression of core circadian genes entrained to a temperature cycle and under free-running conditions. Only *Cry2* was significantly rhythmic (LSP, p<0.05).

The online version of this article includes the following source data and figure supplement(s) for figure 2:

**Source data 1.** Behavioral rhythmicity results.

**Figure supplement 1.** Zeitgeber cycles with a 12 hr period entrain circadian rhythms in *Nematostella*.

repeated the experiment using temperature cycles with a 12 hr period (17–23 °C, 1 degree per hour; *Figure 2—figure supplement 1b*). We found that 10/23 animals had a circadian component and 6/23 had a circatidal component. Once again, after aligning animals within 12 hr windows, the mean behavior profile had a clear 24 hr period and only a weak 12 hr component, suggesting that short-period temperature cycles synchronize circadian rhythms in *Nematostella*.

We used qRT-PCR to measure the expression of several core circadian genes (*Clock*, *Cry1a*, *Cry1b*, *Cry2*, and *Helt*) over 48 hr of the 8–32 °C temperature cycle, and also during free-run. Only *Cry2* was rhythmic during the temperature cycle (LSP, p<0.001), with peak expression at roughly ZT0 (*Figure 2c*). No genes were rhythmic during free-run.

## Aligned light and temperature cycles entrain robust diel rhythms

We tested how the relationship between light and temperature cycles affects circadian behavior by acclimating groups of anemones to a gradually changing temperature cycle (14–26 °C) simultaneously with a 12:12 light-dark cycle. A schematic of the experimental design is shown in *Figure 1b*. When light and temperature cycles were aligned such that lights-on corresponded to the coldest part of the temperature cycle (ZT0), *Nematostella* exhibited clear rhythmic behavior during both entrainment (*Figure 3a*; eJTK, p=2 × 10[−5]) and free-running conditions (eJTK, p=7 × 10[−5]). During the cycles, 20/24 individual (83%) had significant 24 hr rhythms and peak activity occurred at ZT19 (*Table 1*), consistent

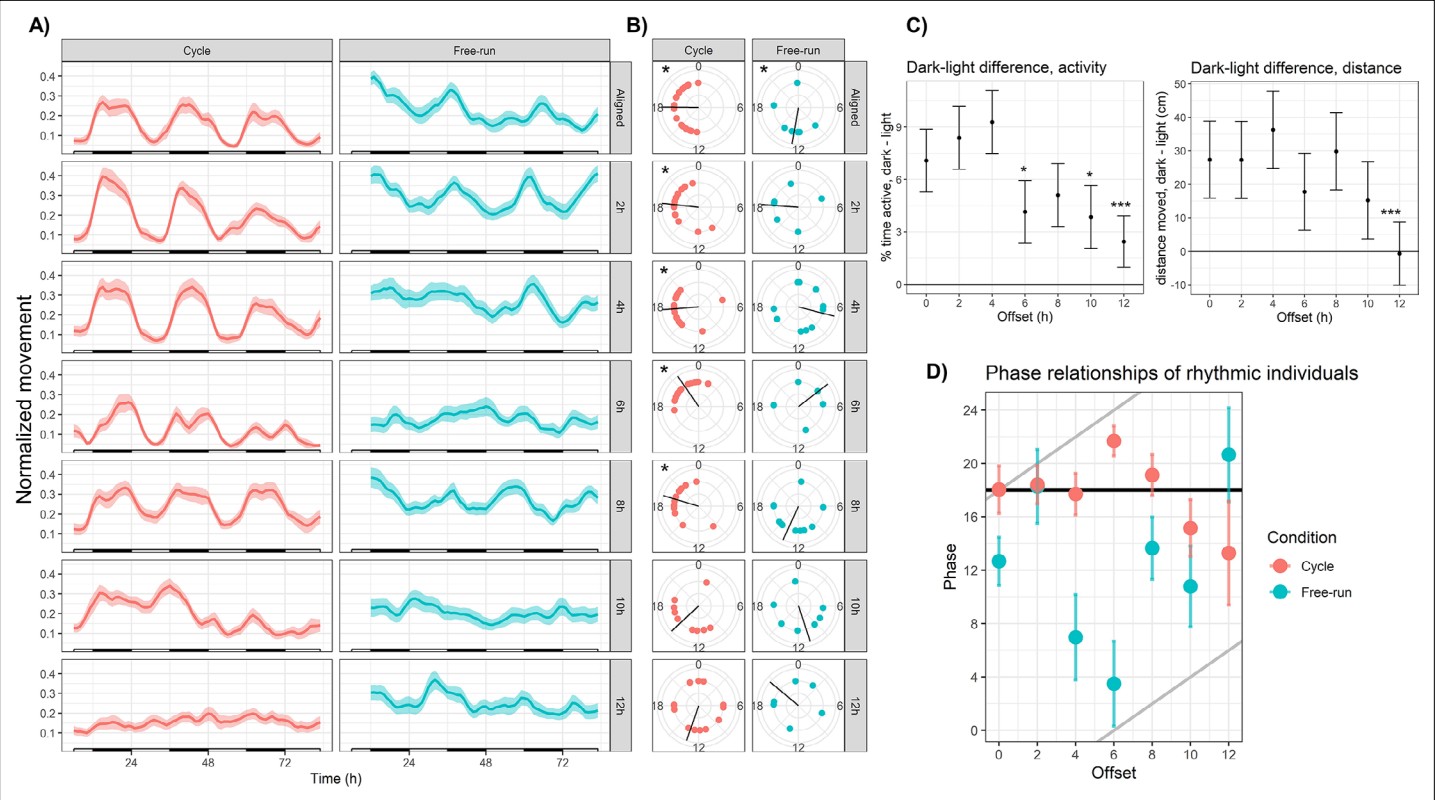

**Figure 3.** Sensory conflict disrupts rhythmic locomotor behavior in *Nematostella*. (**A**) Mean behavior profiles. Individual locomotor profiles were normalized, averaged, and smoothed (see Materials and methods). Shaded area represents standard error. Left, entrainment conditions; white and black bars indicate lights-on and lights-off. Right, free-running at 20 °C and constant darkness; gray and black bars indicate subjective day and night. n=24 for each group except Off12-cycle, where n=36. (**B**) Phases of rhythmic animals calculated by MFourFit. Black line represents circular mean. *:Rayleigh test, p<0.05. (**C**) Difference, in percentage points, between percent time active per hour during dark and light phases (left), and difference between average distance moved per hour during dark and light phases (right). Asterisks indicate significant difference from Aligned-cycle. *: p<0.05; ***: p<0.001. (**D**) Means and standard errors of the phases of rhythmic individuals in each group. Black and grey lines represent expected phases of light-entrained and temperature-entrained rhythms, respectively (see 'Relative strengths of light and temperature zeitgebers').

The online version of this article includes the following figure supplement(s) for figure 3:

**Figure supplement 1.** Period lengths of rhythmic individuals.

**Figure supplement 2.** Lomb-Scargle Periodogram (LSP) power across groups.

with the phases of locomotor activity driven by light (ZT16-20, *Hendricks et al., 2012*; *Oren et al., 2015*) and temperature cycles (ZT17-18, *Figure 2a*). During free-run, 10/24 individuals (42%) had significant 24 hr rhythms and peak activity occurred at ZT12 (*Table 2*).

## Sensory conflict disrupts rhythmic behavior

We tested the effects of long-term entrainment to different light and temperature regimes by delaying the phase of the temperature cycle relative to the light cycle in 2 hr increments (*Figure 1b*). Anemones were transferred from aligned cycles to one of 6 offset regimes (2, 4, 6, 8, 10, and 12 hr offsets) and acclimated for at least two weeks. Within each zeitgeber regime, we recorded the behavior of separate groups of animals during entrainment and free-running conditions, resulting in a total of 14 experimental groups including aligned conditions. In these experiments, ZT0 always refers to lights-on. We use the following notation to refer to experimental groups: Off6-cycle refers to a 6 hr offset between zeitgebers, recorded during the cycles; Off6-FR refers to behavior recorded during free-run. We quantified the strength of rhythmicity using several metrics: the maximum power of Lomb-Scargle periodograms; the amplitude of mean time series; the number of individuals with significant 24 hr rhythms above a strict cutoff (eJTK p<0.001); the relative synchronization (circular variance) of the phases of rhythmic individuals; and the difference in activity between light and dark phases.

**Table 1.** Summary statistics for sensory conflict experiments: Entrainment groups.

| Group | LSP p-value* | LSP power | eJTK p-value† | n rhythmic‡ | Period § | Phase § | Mean phase ¶ | Phase variance** | Amplitude†† |
|---|---|---|---|---|---|---|---|---|---|
| Aligned-cycle | 7.9e-4 | 0.85 | 2e-5 | 20/24 | 23.60 | 19.29 | 18.04 | 0.34 | 0.091 |
| Off2-cycle | 7.9e-4 | 0.75 | 2e-5 | 20/24 | 23.48 | 18.71 | 18.42 | 0.24 | 0.12 |
| Off4-cycle | 7.9e-4 | 0.84 | 2e-5 | 18/24 | 23.76 | 18.90 | 17.70 | 0.27 | 0.12 |
| Off6-cycle | 7.9e-4 | 0.60 | 2e-5 | 16/24 | 24.30 | 20.16 | 21.68 | 0.15 | 0.075 |
| Off8-cycle | 7.9e-4 | 0.85 | 2e-5 | 15/24 | 23.60 | 19.20 | 19.12 | 0.26 | 0.085 |
| Off10-cycle | 3.6e-3 | 0.19 | 1.7e-4 | 10/24 | 22.60 | 17.75 | 15.15 | 0.45 | 0.042 ‡‡ |
| Off12-cycle | 7.9e-4 | 0.22 | 4.5e-3 | 13/36 | 23.66 | 18.62 | 13.29 | 0.86 | 0.015 ‡‡ |

*Tested periods between 20–28 hr.
†Tested period of 24 hr.
‡ eJTK $p < 1 \times 10^{-3}$.
§determined by MFourFit.
¶Circular mean of the phases of rhythmic individuals (eJTK $p < 1 \times 10^{-3}$).
** Circular variance of the phases of rhythmic individuals (eJTK $p < 1 \times 10^{-3}$).
††Determined with CircaCompare.
‡‡:Reduced amplitude compared to Aligned-cycle, CircaCompare, $p < 0.05$. LSP: Lomb-Scargle periodogram.

Large degrees of misalignment profoundly disrupted rhythmic behavior. During the largest possible misalignment (Off12-cycle) the normal activity pattern was visibly disrupted (*Figure 3a*) and 24 hr rhythmicity was only weakly detectable in the mean behavior profile (eJTK, $p = 4.5 \times 10^{-3}$; *Table 1*). The power and amplitude of the mean time series were lower than Aligned-cycle (*Table 1*; Dunn test, $p < 0.05$; CircaCompare, $p < 0.05$), and the difference in activity between dark and light phases was also severely reduced in terms of both time active (linear mixed effects model; LMM, $p = 1 \times 10^{-4}$) and distance moved (LMM, $p = 2 \times 10^{-4}$; *Figure 3c*). Although 13/36 individuals (36%) were rhythmic (eJTK, $p < 0.001$), their phases could not be distinguished from a circular uniform distribution (Rayleigh test, $p = 0.8$), indicating desynchronization. Behavioral rhythms were also visibly disrupted and quantitatively weaker in Off10-cycle compared in Aligned-cycle (*Figure 3a*, *Table 1*), including reduced power (Dunn test, $p < 0.05$), amplitude (CircaCompare, $p < 0.05$), and dark:light activity ratio (LMM $p = 0.013$).

**Table 2.** Summary statistics for sensory conflict experiments: Free-running groups.

| Group | LSP p-value* | LSP power | eJTK p-value† | n rhythmic‡ | Period § | Phase § | Mean phase ¶ | Phase variance** | Amplitude†† |
|---|---|---|---|---|---|---|---|---|---|
| Aligned-FR | 7.9e-4 | 0.47 | 7e-5 | 10/24 | 25.84 | 11.63 | 12.67 | 0.35 | 0.060 |
| Off2-FR | 7.9e-4 | 0.79 | 7e-5 | 6/24 | 22.62 | 16.50 | 18.28 | 0.63 | 0.071 |
| Off4-FR | 7.9e-4 | 0.51 | 7e-5 | 11/24 | 23.64 | 14.17 | 6.98 | 0.74 | 0.050 |
| Off6-FR | 0.13 | 0.095 | 0.76 | 5/24 | 27.34 | 18.57 | 3.50 | 0.73 | 4.9e-3 ‡‡ |
| Off8-FR | 7.9e-4 | 0.67 | 7e-5 | 11/24 | 21.74 | 15.53 | 13.65 | 0.51 | 0.048 |
| Off10-FR | 0.53 | 0.042 | 0.61 | 7/24 | 28.00 | 3.39 | 10.78 | 0.70 | 2.8e-3 ‡‡ |
| Off12-FR | 7.9e-4 | 0.32 | 7e-5 | 6/24 | 22.96 | 11.72 | 20.65 | 0.80 | 0.034 ‡‡ |

*Tested periods between 20–28 hr.
†Tested period of 24 hr.
‡ eJTK $p < 1 \times 10^{-3}$.
§determined by MFourFit.
¶Circular mean of the phases of rhythmic individuals (eJTK $p < 1 \times 10^{-3}$).
**Circular variance of the phases of rhythmic individuals (eJTK $p < 1 \times 10^{-3}$).
††Determined with CircaCompare.
‡‡Reduced amplitude compared to Aligned-FR, CircaCompare, $p < 0.05$. LSP: Lomb-Scargle periodogram.

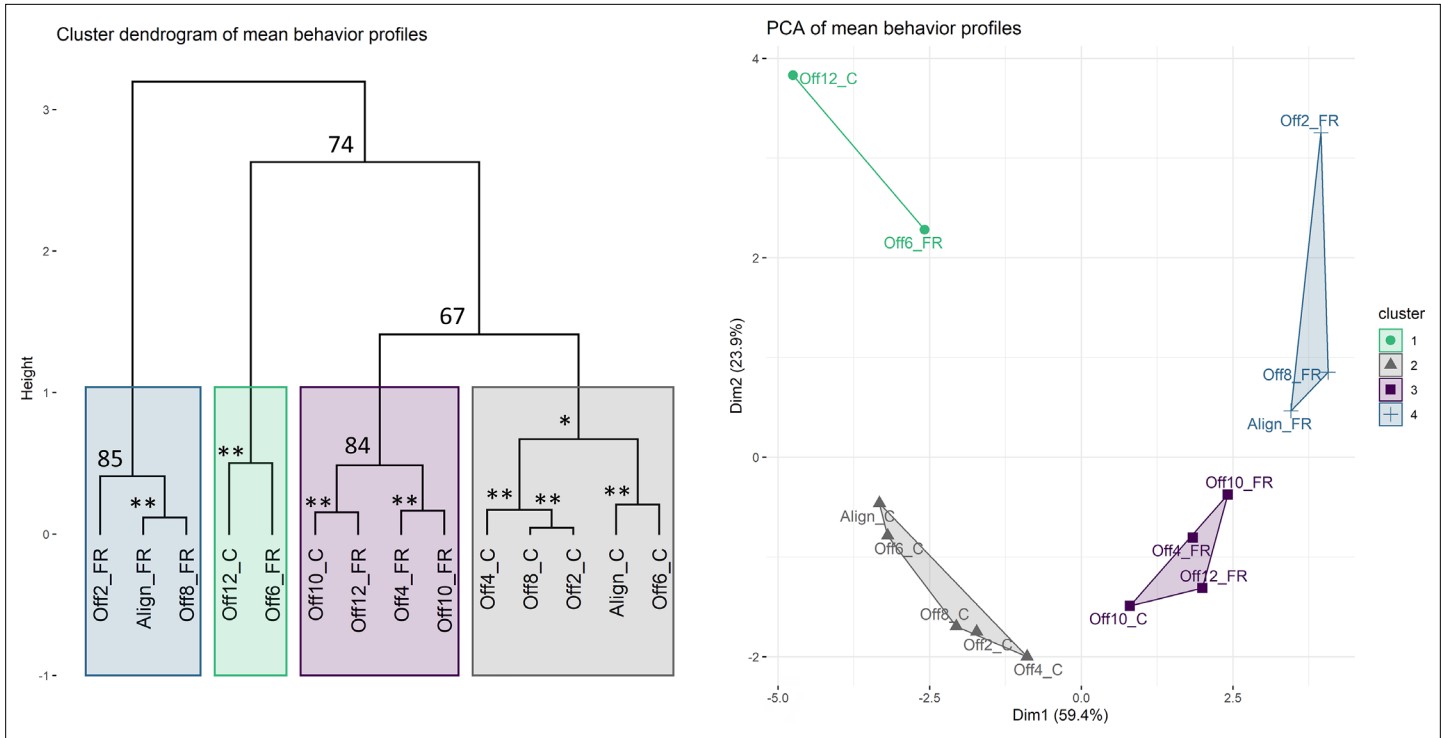

**Figure 4.** Clustering analysis of mean behavior profiles based on wavelet transformation. (**A**) Principal component analysis of wavelet transformation distance matrix. Time series were grouped and colored based on the clusters identified in (**B**). C: cycles; FR: free-running. (**B**) Hierarchical clustering of samples in principal components space. Colored rectangles indicate the four clusters discussed in the text. Numbers indicate unbiased (AU) p-values. *AU p-value $\geq$ 90; **AU p-value $\geq$ 95.

The phase of the mean time series in Off10-cycle was also advanced by 1.5 hr (CircaCompare, p=5 × $10^{-5}$). The reduced dark-light activity ratios during Off10-cycle and Off12-cycle were specifically due to increased activity during lights-on (LMM, p<0.05), with no difference in activity during lights-off (LMM, p>0.1).

Weakened locotomor rhythms in Off10-cycle and Off12-cycle were caused both by a weakening of the rhythms of individual animals, and a lack of synchronization between the remaining rhythmic individuals. The percentage of rhythmic individuals declined monotonically with increasing misalignment, from 83% in Aligned-cycle to 36% in Off12-cycle (*Table 1*), and the LSP powers of animals in Off10-cycle and Off12-cycle were lower than during Aligned-cycle (Dunn test, $p < 0.05$; *Figure 3—figure supplement 2*). In addition, the phases of rhythmic individuals in Off10-cycle and Off12-cycle had much larger variances than other groups and could not be distinguished from uniform distributions (*Table 1*; Rayleigh test, p>0.05). Thus, under severe sensory conflict (SC), individual *Nematostella* were less likely to exhibit rhythmic behavior, and less likely for that behavior to synchronize with other individuals.

Anemones exhibited robust diel behavior under smaller degrees of misalignment (0–8 hr; *Figure 3*). The Off2-cycle, Off4-cycle, and Off8-cycle groups exhibited rhythms similar to those in Aligned-cycle, with no significant phase shifts or reductions in rhythm strength. The Off6-cycle group also showed 24 hr rhythms, but the mean phase of rhythmic individuals was delayed by 3.6 hr relative to Aligned-cycle (Watson test, p=0.0046); the mean behavior profile was also delayed (CircaCompare, p=4 × $10^{-5}$). The dark:light activity ratio was reduced (LMM, p=0.024) in Off6-cycle, and there was a marginally significant circatidal (10–14 hr) component (LSP, p=0.006). We confirmed that *Nematostella* exhibited normal diel behavior under a broad range of zeitgeber regimes using clustering analysis. We quantified distances between the 14 mean time series based on their spectral properties using wavelet transformation followed by principal component analysis and hierarchical clustering (see Materials and methods). The Aligned-cycle, Off2-cycle, Off4-cycle, Off6-cycle, and Off8-cycle groups formed a single cluster with high support based on the approximately unbiased (AU) test, corresponding to normal rhythmic behavior, whereas Off10-cycle and Off12-cycle fell into different clusters (*Figure 4*).

*Nematostella*'s diel behavior was therefore robust up to an 8 hr misalignment between light and temperature, although there was a detectable phase delay specifically in Off6-cycle.

## Sensory conflict disrupts endogenous rhythms

The above disruptions of behavior during SC could be due to masking effects rather than disruption of endogenous rhythms. However, we found that free-running behavior was also disrupted by SC, demonstrating *bona fide* effects on the endogenous clock (*Tables 1 and 2*, *Figure 3a*). The mean behavior of animals in Off6-FR and Off10-FR was totally arrhythmic (LSP and eJTK, p>0.1). Mean behavior in Off12-FR was rhythmic (eJTK, p=7 × 10⁻⁵), but the power and amplitude of the mean time series were reduced compared to Aligned-FR (Dunn test, p<0.05; CircaCompare, p<0.05), there were fewer rhythmic individuals, and the variance of the phases of rhythmic individuals was larger (*Table 2*). In the clustering analysis, Aligned-FR, Off2-FR, and Off8-FR formed one cluster with moderate AU support, while Off4-FR, Off10-FR, and Off12-FR formed a cluster that also included Off10-cycle (*Figure 4*). The final cluster was strongly supported and consisted of Off6-FR and Off12-cycle, two groups with severely disrupted rhythms.

Behavioral patterns during free-running differed from patterns during entrainment conditions. In fact, Off12-FR was more strongly rhythmic than Off12-cycle (*Table 2*). Thus, antiphasic zeitgeber cycles weakened, but did not abolish, endogenous circadian rhythms, and behavior was further disrupted during entrainment conditions. Conversely, while endogenous rhythms were severely disrupted during Off6-FR, light and temperature cycles were able to drive rhythmic behavior during Off6-cycle (albeit with quantitative differences from Aligned-cycle). Additionally, disruptions of free-running rhythms did not increase linearly with the degree of offset. Behavior in Off6-FR was arrhythmic, but Off4-FR and Off8-FR exhibited clear rhythmicity (*Table 2*).

Disruption of free-running behavior was due primarily to a lack of synchronization among rhythmic animals, rather than weakened rhythms of individual animals. Only Aligned-FR had a significant non-uniform phase distribution (Rayleigh test, p<0.05). Mean powers of behavioral rhythms did not differ across free-running groups (Dunn test, p>0.05), nor was there a clear trend in the number of rhythmic individuals (*Table 2*). Phases of free-running rhythms differed in some groups, but not according to an obvious pattern. The phase of Off2-FR was advanced by 1.8 hr relative to Aligned-FR, Off8-FR was advanced by 4.4 hr, and Off12-FR was advanced by 4.3 hr (CircaCompare, p<7 × 10⁻³). Period length was not affected by SC (*Figure 3—figure supplement 1*).

## Relative strengths of light and temperature cycles

To gain insight into the relative influence of light and temperature on behavioral rhythms, we compared the phases of rhythmic individuals against the offset between light and temperature (*Figure 3d*). In isolation, temperature and light each drive rhythms that peak at roughly ZT18 (*Figure 2a*; *Hendricks et al., 2012*; *Oren et al., 2015*), which we plot as the black and grey lines in (*Figure 3d*), respectively. If the phase of a rhythm were determined entirely by light, it would be close to the black line; if determined by temperature, it would be close to the grey lines. The phases of Off4-cycle and Off8-cycle groups were significantly closer to the light-entrained (black) line (paired Wilcoxon rank tests, p<0.05). All other groups were approximately equidistant between zeitgebers, including every free-running group. This illustrates that light exerts strong direct control on behavior, although light and temperature interact to set the phase of circadian behavior. When cycles were close in phase (Aligned and Off2), they acted together the set the phase of locomotor rhythms, while at larger offsets phases were either determined primarily by the light cycle, or were intermediate between light and temperature. On the other hand, we found no evidence that the phases of free-running rhythms were primarily determined by either light or temperature, with the caveat that this test had less power for free-running groups due to the numbers of rhythmic animals. Thus, although light has strong direct effects on behavior, temperature also influences the phase of entrained rhythms.

## Sensory conflict alters transcriptome-wide patterns of rhythmic gene expression

We used 3' tag-based RNA sequencing (Tag-seq) to characterize the effects of SC on gene expression. Anemones were sampled at 13 time points across 48 hr, either under Aligned-cycle or Off12-cycle

conditions (*Figure 1b*). Reads were mapped to the SimRbase genome (Nvec200_v1, https://genomes.stowers.org/starletseaanemone).

SC substantially altered, and in many cases inverted, diel gene expression patterns in *Nematostella*. Under Aligned conditions, 1009 genes (6.6% of the transcriptome) were differentially expressed (DE) between light and dark samples (p<0.05), while 627 genes were DE between light and dark during SC, and 277 genes were DE in both SC and aligned light and temperature cycles. The correlation of log-fold changes of the 277 shared genes was negative (Pearson's r=-0.59), meaning that genes upregulated in the light under Aligned conditions were generally downregulated in light under SC, and vice versa. Consequently, only 25 genes were DE between light and dark samples when averaged across both time series, and 1747 genes differed in their response to light across conditions. Using discriminant analysis of principal components (DAPC), we confirmed that genes distinguishing light and dark samples under Aligned conditions failed to do so during SC (*Figure 5a*). When light and dark SC samples were plotted on the discrimnant axis that maximized the variance between light and dark Aligned samples, the light SC animals were closer to the dark Aligned animals, showing that the diel expression of many genes was reversed.

Temperature was a stronger regulator of gene expression than light in our experiment: 1978 genes (13%) had a significant linear response to temperature, compared to only 25 DE genes in response to light. However, light-responsive genes notably included three putative circadian transcription factors (*PAR-bZIP-a*, *PAR-bZIP-c*, and *Helt*). Temperature-responsive genes were enriched for gene ontology (GO) terms related to transcription factor binding and oxidoreductase activity; genes with a significant condition:light interaction were also enriched for terms related to transcription factor activity. This suggests that the temperature cycle had widespread effects on transcription during light-dark cycles and modulated the expression of transcription factors. The mean expression level of 90 genes significantly differed between Aligned and SC time series. These genes were not enriched for any GO terms, but included one of the aforementioned PAR-bZIP proteins and a cytochrome P450 oxidase (CYP) previously implicated in circadian rhythms in *Nematostella* (*Oren et al., 2015*), along with two other CYPs and a SOX-family transcription factor (*Figure 5—source data 2*). Many genes (1898) displayed a linear change in expression with time, presumably because animals were not fed during the experiment. Genes with decreasing expression over time were enriched for GO terms related to cellular metabolism, suggesting an overall downregulation of metabolic processes as animals went without food. For complete lists of DE genes and GO enrichment results, see *Figure 5—source data 2* and *Figure 5—source data 4*.

We searched 1 kb upstream of transcription start sites ('promoters') for putative binding motifs that may play a role in regulating circadian gene expression using the HOMER motif discovery tool. Genes whose mean expression during SC differed from mean expression during Aligned conditions were enriched for a motif of unknown function, which was most similar to Myb-related binding motifs in plants (HOMER, $p=1 \times 10^{-12}$; *Figure 5—source data 3*), while genes with a negative condition:light interaction (i.e. genes with relatively more dark expression during SC) were strongly enriched for numerous bZIP transcription factor binding motifs (HOMER, $p=1 \times 10^{-14}$). Genes whose expression was negatively associated with temperature, and genes whose expression was positively associated with time, were also enriched for bZIP motifs (*Figure 5—source data 3*).

## Rhythmicity analysis

We tested for rhythmic gene expression using RAIN (rhythmicity analysis incorporating nonparametric methods) v1.24.0 (*Thaben and Westermark, 2014*), a non-parametric approach particularly suited for the detection of asymmetric waveforms. We compared RAIN to two other pieces of software: ECHO (*De Los Santos et al., 2020*), a parametric method based on the harmonic oscillator equation, and DryR (*Weger et al., 2021*), another parametric approach based on harmonic regression. Encouragingly, there was strong agreement between the three programs despite their completely different statistical methodologies (see Appendix 1). We focus on the output from RAIN, and the output of the other two programs is available in *Figure 5—source data 2*.

Surprisingly, there were similar overall numbers of rhythmic genes in Aligned and SC conditions, although less than half of them were rhythmic in both time series. Under Aligned conditions, 2868 genes displayed 24 hr rhythms in expression (RAIN, p<0.01; 18% of the transcriptome), while 2440 genes were rhythmic during SC, and 1116 genes were shared (*Figure 5b*). To define condition-specific

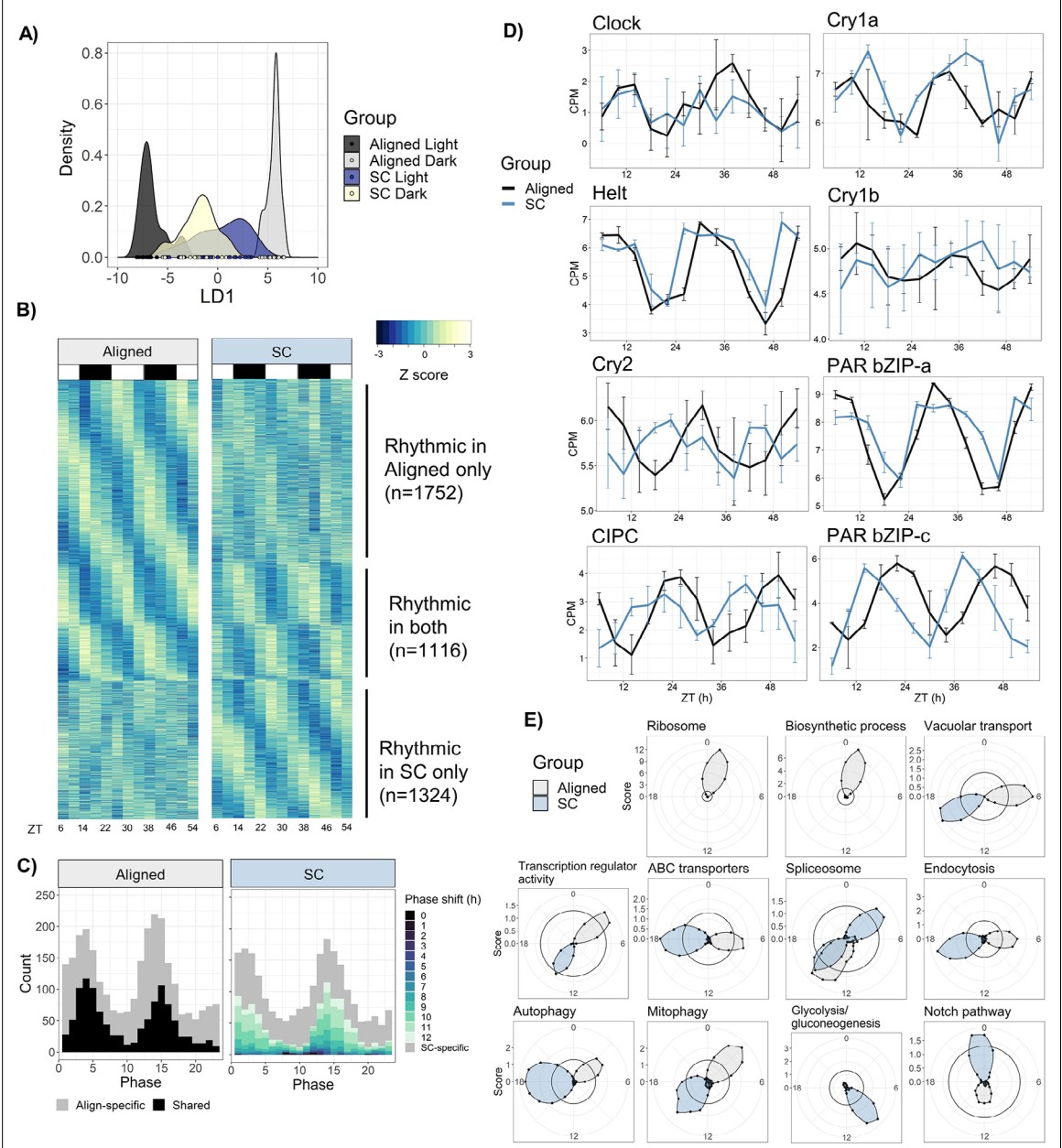

**Figure 5.** Sensory conflict alters patterns of rhythmic gene expression. (**A**) Discriminant analysis of principal components (DAPC) demonstrates shifts in diel gene expression under SC. Density plot of sample loading values of light and dark samples during Aligned and SC conditions, on 1st discriminant axis from DAPC of aligned samples ('LD1'). (**B**) Heatmaps of normalized rhythmic gene expression. Replicates were averaged at each time point and Z-scores were calculated for each gene and time series. 'Rhythmic' genes had a RAIN p-value < 0.01 in the Aligned time series only (top), SC time series only (bottom), or both (middle). White and black rectangles represent light and dark time points, respectively. (**C**) Phase distributions of rhythmic genes in Aligned (left) and SC (right) time series. Grey genes were rhythmic in only one time series, and black and colored genes were rhythmic in both. In SC, the color of shared rhythmic genes represents the phase shift of that gene from the Aligned time series. (**D**) Phases of core clock genes under Aligned and SC conditions. Lines show mean counts per million, and error bars represent 95% confidence intervals. (**E**) Sliding window enrichment analysis of select GO and KEGG terms during Aligned and SC time series. P-values were calculated by comparing genes with peak phase within a 4 hr sliding window with all other genes, and the score (y-axis) at each point is the average adjusted -log10 p-value at that time point (see Methods). Black circles indicates an FDR threshold of 0.05.

The online version of this article includes the following source data and figure supplement(s) for figure 5:

**Source data 1.** Counts matrix.

**Source data 2.** Differential expression, rhythmicity, and WGCNA analyses, and gene annotations.

*Figure 5 continued on next page*

*Figure 5 continued*

**Source data 3.** Results of HOMER motif enrichment analyses.

**Source data 4.** GO and KEGG enrichment results.

**Figure supplement 1.** Amplitude and mean expression of rhythmic genes.

rhythmic genes, we excluded genes that were marginally significant in the other condition (p<0.1); by this metric, 1225 genes lost rhythmicity during SC (Align-specific) and 921 gained rhythmicity (SC-specific). The promoters of shared rhythmic genes were enriched for bZIP transcription factor binding sites (p=1 × 10$^{-6}$), while the promoters of Align-specific and SC-specific rhythmic genes were not enriched for any motifs (*Figure 5—source data 3*).

To understand how functional categories of rhythmic genes were affected by SC, we used a sliding window approach for functional enrichment of rhythmic genes (see Materials and methods). This revealed dramatic changes to the expression of genes that mediate macromolecule, protein, and RNA metabolism during SC (*Figure 5e*). Most terms that were enriched among aligned rhythmic genes were not enriched at any time during SC, and only a single GO or KEGG term, 'Spliceosome', was enriched at the same time of day in both groups. Under aligned conditions, terms related to macromolecule and protein metabolism were enriched in early morning (ZT0-3) and lost enrichment under SC, while 'RNA transport' and 'ubiquitin-mediated proteolysis' were enriched in early night (ZT13-14) and lost rhythmicity under SC. Other terms shifted from day to night during SC, including 'transcription factor activity', 'vacuolar transport', 'hydrolase activity', 'autophagy', 'mitophagy', 'ABC transporters', and 'mRNA surveillance pathway'; 'protein processing in ER' shifted from ZT10 to ZT4 (*Figure 5—source data 4*). Some terms were only enriched under SC, including 'DNA binding', 'Notch signaling pathway', 'ATP binding', 'GTPase activity', 'Glycolysis', and 'Pyruvate metabolism' (*Figure 5—source data 4*). For full enrichment results, see *Figure 5—source data 4*.

The phase distribution of Aligned rhythmic genes was bimodal (Rayleigh test, p=1 × 10$^{-4}$), with peaks in mid-morning (ZT4) and early night (ZT14) (*Figure 5c*). Genes with phases during the two peaks (ZT0-6 and ZT12-18) were less likely to lose rhythmicity during SC (38% lost rhythmicity compared to 53% of other genes; chi-squared test, p<2 × 10$^{-16}$), suggesting that rhythmic gene expression during these windows was somehow more robust. The phase distribution under SC was also bimodal (*Figure 5c*), although the two distributions were not identical (Rao's test for circular homogeneity, p=0.046). The vast majority of genes that were rhythmic under both Aligned and SC conditions (1092/1116, 98%) shifted in phase relative to the light cycle, with a median phase shift of 10.3 hr (CircaCompare, p<0.01); 86% shifted by at least 6 hr, and 58% by 10–12 hr. Only 24 genes did not shift in phase relative to the light cycle, including *Clock*, *Helt*, and *PAR-bZIP-a* (*Figure 5b*). We identified 647 genes that shifted in phase by 10–12 hr relative to the light cycle and did not shift in phase relative to the temperature cycle; these genes closely tracked the phase of the temperature cycle and may be directly regulated by temperature. They were enriched for GO terms related to transcription factor activity, and the KEGG terms 'Mitophagy' and 'Spliceosome' (*Figure 5—source data 4*).

**Table 3.** Comparison of phases of putative clock genes.

| Study | Clock | Helt | CIPC | PAR-bZIP-a | PAR-bZIP-c | Cry1a | Cry1b | Cry2 |
|---|---|---|---|---|---|---|---|---|
| Current study (Aligned) | 12.4 | 9.2 | 1.0 | 7.3 | 22.3 | 9.3 | 10.5 | 5.6 |
| Current study (SC) | 11.94 | 7.9 | 19.3 | 8.0 | 16.2 | 12.1 | NA | 21.8 |
| *Oren et al., 2015*\* | 10.9 | 7.3 | 23.4 | 5.3 | 19.4 | 7.9 | NA | 2.3 |
| SimrBase ID | 6258 | 15017 | 4651 | 8136 | 8448 | 8109 | 8041 | 15282 |
| JGI ID | 160110 | 246249 | 245026 | 150375/ 87565 | 39846 | 168581 | 106062 | 194898 |

Phases estimated with CircaCompare. NA: below significance cutoff, phase not calculated. We used a p-value cutoff of p=0.01 for the current study, and p=0.05 for (*Oren et al., 2015*) because of their small sample size.

The amplitude and mean expression of rhythmic genes were slightly lower overall during SC. Mean expression of SC-specific genes was 10% lower than that of Align-specific genes during their respective time series (Wilcoxon, p=1 × 10$^{-8}$; *Figure 5—figure supplement 1*), but expression levels of shared rhythmic genes did not differ during SC (p=0.9). The amplitudes of Align-specific and SC-specific genes did not differ (p=0.49), but the amplitudes of shared rhythmic genes (n=1116) were median 7.1% lower during SC (Wilcoxon, p=2 × 10$^{-4}$; *Figure 5—figure supplement 1*). Specifically, amplitudes of the 647 temperature-responsive genes were reduced by median 8.8% (Wilcoxon, p=7 × 10$^{-4}$).

## Comparison with light cycles at constant temperature

To assess how rhythmic gene expression during simultaneous light and temperature cycles compares to gene expression during a light cycle at constant temperature, we re-analyzed a previous study that sampled *Nematostella* during an LD cycle at 23 °C (*Oren et al., 2015*). The phases of clock genes between *Oren et al., 2015* and the current study are shown in *Table 3*, and this comparison is discussed in more detail in Appendix 2. The phases of core circadian genes (*Clock*, *Helt*, *CIPC*, cryptochromes, and PAR-bZIPs) were within 4 hr between our Aligned time series and *Oren et al., 2015*, with the exception of *Cry1b*. However, there was little overlap between rhythmic genes overall, as only 477/1498 (32%) genes rhythmic in *Oren et al., 2015* (RAIN, p<0.05) had a p-value less than 0.05 in the current study. Differences in clock output may be due to experimental conditions or differences between populations. Nonetheless, our re-analysis shows that core clock gene expression was not altered by the addition of a temperature cycle in-phase with the light cycle.

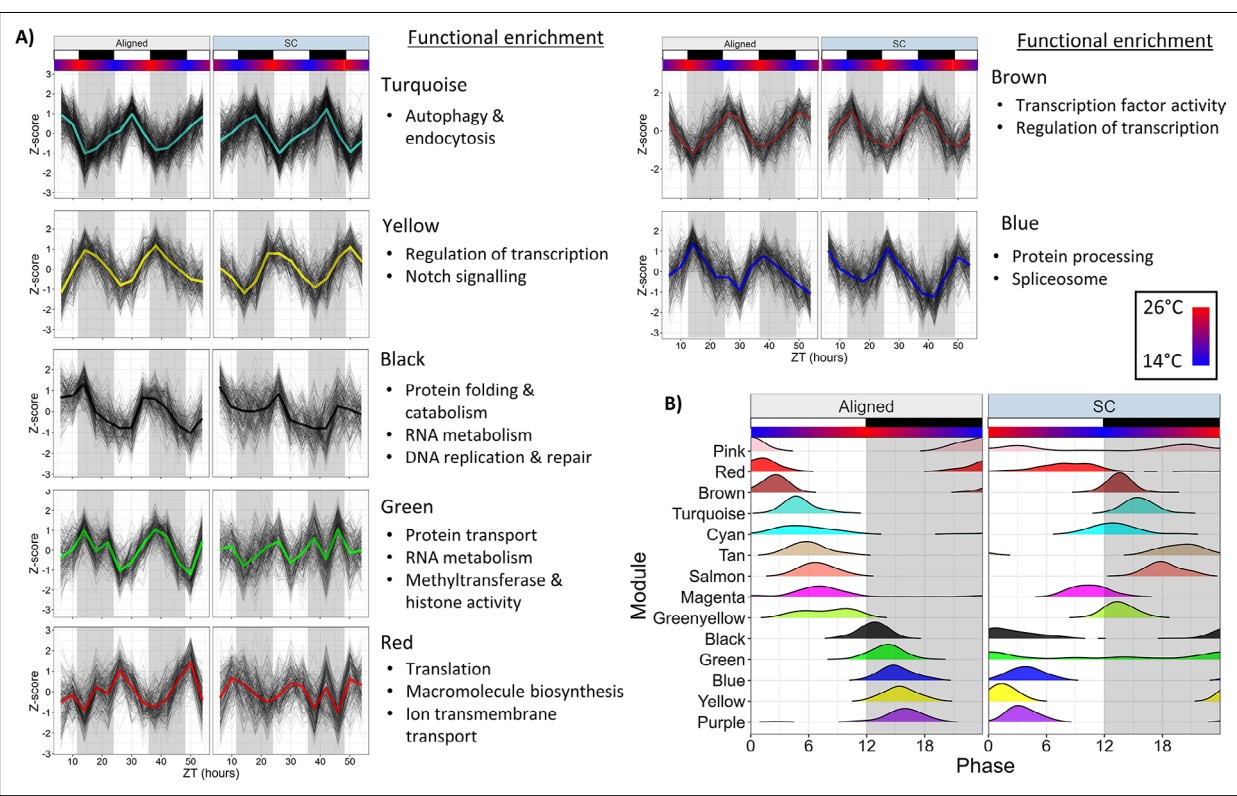

**Figure 6.** Network analysis identifies changes to rhythmically co-expressed gene modules. (**A**) Expression patterns of select modules in the full network. Black lines are Z-scores (difference from mean divided by standard deviation) of the expression of each gene, and the colored line is the mean of those Z-scores. Left panels are Aligned samples, and right panels are SC samples. (**B**) Phase distributions of modules genes in the Full network during Aligned (left) and SC (right) conditions. Each line shows a smoothed density estimate of the phases of rhythmic genes within that module, weighted by the -log10(p-value) from RAIN. ZT0 refers to lights-on. White and black boxes indicate lights-on and lights-off, respectively; colored bars indicate temperature.

The online version of this article includes the following figure supplement(s) for figure 6:

**Figure supplement 1.** WGCNA module expression, Full network.

## Network analysis

We used weighted gene co-expression network analysis (WGCNA) to characterize co-expressed modules of rhythmic genes. We constructed a network using all genes rhythmic in either time series (n=4192 genes), which were assigned to 14 modules. The expression of select modules is shown in *Figure 6a*, all modules are shown in *Figure 6—figure supplement 1*, and the timing of module expression is shown in *Figure 6b*. Functional enrichment results are available in *Figure 5—source data 4*.

Co-expression patterns of rhythmic genes were substantially weakened during SC. Intramodular connectivity (kIM) of all genes was reduced by 23% during SC, and the connectivity of each gene to all other genes in the network (kTotal) was reduced by 12% (Wilcoxon, $p<1 \times 10^{-15}$). The correlation between genes and their respective eigengene (kME; eigengene is the first principal component of the module expression matrix) was reduced by 5.6% (Wilcoxon, $p=6 \times 10^{-15}$), and the proportion of each gene's expression variance explained by eigengene expression (propVar) was reduced by 8.2% (Wilcoxon, $p=1 \times 10^{-11}$). We also calculated these connectivity metrics for two networks constructed separately for Aligned and SC samples; kIM was 43% lower in the SC network and kTotal was 31% lower (Wilcoxon, $p<1 \times 10^{-15}$). These results show that the expression of rhythmic genes was less well-correlated both within and across network modules during SC.

Distinct functionally-related groups of genes responded to SC in different ways. Half (7/14) of the module eigengenes were strongly rhythmic in both Aligned and SC condition (LSP, p<0.01) and shifted in phase by 10–12 hr in accordance with the temperature cycle, again illustrating that temperature was a prominent driver of rhythmic gene expression. Four of these modules peaked in expression in the light phase during aligned conditions and shifted to dark-phase expression during SC. These modules were enriched for terms related to autophagy and endocytosis (Turquoise); transcription factor activity (Brown); ATP hydrolysis and ABC transporter activity (Tan); and serine-type endopeptidas activity (Salmon). Promoters of Turquoise and Brown modules were enriched for bZIP binding motifs (HOMER, $p<1 \times 10^{-8}$), and Salmon module promoters were enriched for ETS-binding motifs (HOMER, $p=1 \times 10^{-5}$). Modules that peaked in expression during the dark phase during aligned conditions and shifted to light-phase expression during SC were enriched for terms related to protein and RNA metabolism, chromatin organization, and DNA replication and repair (Black); spliceosome components, protein processing, and butanoate metabolism (Blue); and regulation of transcription, protein dimerization activity, and Notch signaling (Yellow). Modules that lost rhythmicity during SC were enriched for terms related to RNA metabolism, protein transport, and regulation of gene expression (Green); and sphingolipid metabolism (Pink). Green module promoters were enriched for putative binding motifs with unknown function (HOMER, $p=1 \times 10^{-10}$), and this module contained the *Clock* gene. The Red module advanced in phase by 8 hr and was only marginally rhythmic during SC (LSP, p=0.019), and was enriched for terms related to proton transmembrane transport and translation. A module that gained rhythmicity during SC was enriched for genes related to thiamine and purine metabolism (Purple), and its promoters were enriched for helix-turn-helix binding motifs (HOMER, $p=1 \times 10^{-5}$). Finally, two modules were marginally rhythmic in Aligned conditions (LSP, p=0.02) and shifted in phase by 4–8 hr during SC. The Cyan module was enriched for terms related to glycolysis, but this enrichment was driven only by two genes, both of which were annotated as phosphoenolpyruvate carboxykinases *Figure 5—source data 2*; Cyan module promoters were enriched for SOX binding motifs (HOMER, $p=1 \times 10^{-10}$). The Magenta module was enriched for terms related to protein catabolism, oxidoreductase activity, and RNA transport.

## Discussion

We show here that ecologically relevant temperature cycles drive rhythmic behavior and influence circadian rhythms in the cnidarian *Nematostella vectensis*, and explore how the relationship between simultaneous light and temperature cycles affects behavior and gene expression. Given that white light exerts strong direct effects on behavior in *Nematostella*, including masking free-running locomotor rhythms (*Oren et al., 2015*; *Tarrant et al., 2019*), we thought behavior might simply synchronize to the light cycle to the exclusion of temperature. However, *Nematostella*'s behavior became severely disrupted and arrhythmic on average when light and temperature cycles were offset by 10–12 hr (*Figure 3a*). Free-running rhythms were also disrupted, depending on the specific phase

relationship. This shows that both light and temperature interact to set the phase of *Nematostella*'s clock, and that neither cue dominates the other. As a consequence, normal rhythmic behavior is only possible under certain relationships between environmental signals. The disruptive effects of sensory conflict (SC) were not acute responses to transient conditions, but chronic changes measured after weeks of exposure and acclimation.

Gene expression rhythms were also dramatically altered by SC (*Figure 5b*). Surprisingly, few genes remained in-phase with the light cycle across different temperature cycles, whereas several hundred genes followed the temperature cycle regardless of light. Rather than simply reducing the number of rhythmic transcripts, SC caused turnover in the identities of rhythmic genes and altered the relative timing of expression between them. This resulted in widespread perturbation of the temporal expression of metabolic processes and an overall weakening of rhythmic co-expression patterns. Nonetheless, many genes were robustly rhythmic during SC, and other genes even gained rhythmic expression. These patterns differed between functional categories. For instance, genes related to autophagy and endocytosis shifted in phase following the temperature cycle, while genes related to ribosome and protein metabolism lost rhythmicity (*Figure 5e*).

## Simultaneous light and temperature cycles produce complex behavior

Rhythmic behavior is the net result of endogenous and exogenous signals, leading to potentially complex behavioral outcomes. Direct effects of environmental cues can mask endogenous rhythms (*Aschoff, 1960*), and, conversely, circadian rhythms can constrain or 'gate' acute environmental responses (*Salter et al., 2003*; *Fowler et al., 2005*). During simultaneous light and temperature cycles, we found that behavior was increasingly disrupted up to the largest degrees of misalignment (10 hr and 12 hr offsets; *Figure 3a*). However, behavioral patterns during entrainment conditions did not directly mirror the corresponding free-running behavior, indicating that direct effects of light and temperature interacted with and masked endogenous behavior. Our data illustrate two contrasting scenarios for the relationship between endogenous and exogenous cues. During the 6 hr offset, endogenous rhythms were strongly disrupted, but behavior during entrainment conditions was roughly normal. In this case, light and temperature directly drove rhythmic behavior, although the underlying clock output was arrhythmic. On the other hand, during the 12 hr offset, endogenous rhythms were relatively robust, but behavior during entrainment conditions was arrhythmic. In this case, misaligned light and temperature cycles masked endogenous rhythms.

Studies of *Drosophila* have noted similarly complex patterns of behavior depending on the relationship between light and temperature cycles. Rhythmic locomotor behavior and molecular oscillations in the fly central clock were disrupted at intermediate (5–7 hr) offsets, but the phase of behavior followed either light or temperature at smaller and larger offsets (*Harper et al., 2016*). In contrast, peripheral clock oscillations appeared to preferentially entrain to light over temperature (*Harper et al., 2017*). *Nematostella* lacks this central-peripheral distinction. In a sense, *Nematostella*'s circadian behavior was more similar to the central clock outputs of *Drosophila* than to peripheral clock oscillations, because rhythmic behavior was disrupted under certain offsets instead of following a single zeitgeber. We also observed prominent non-linear dynamics in free-running behavior, with phase differences of as little as two hours between light and temperature resulting in completely different behavioral outcomes. Free-running rhythms were most strongly disrupted at the 6 hr and 10 hr offsets, and somewhat weakened at the 12 hr offset (*Figure 3a*). However, rhythms were robust at the 4 hr and 8 hr offsets.

The underlying reasons for these specific behavior patterns depend on how the circadian clock receives and processes information from light and temperature. The bilaterian central clock is physically located in the brain and consists of multiple neuronal subpopulations (*Dibner et al., 2010*). In *Drosophila*, some neurons preferentially entrain to temperature cycles, while other neurons only entrain to light cycles (*Miyasako et al., 2007*). Recent work has identified additional subsets of neurons that act as thermosensors, and current data support a model in which separate sets of clock neurons receive warm and cold information; these neurons then integrate temperature with additional cues to regulate clock outputs (*Alpert et al., 2022*). Although the organization of cnidarian clocks probably fundamentally differs from that of bilaterians due their lack of a central nervous system, it is likely that *Nematostella* has neurons dedicated to specific sensory tasks; other neurons may act as clock neurons and coordinate rhythmic gene expression in other cells. Indeed, single-cell studies show that

*Nematostella* has a diverse population of neurons with distinct transcriptional profiles; these neuronal subtypes differ in their expression of clock genes and genes involved in sensory processing, such as cryptochromes and opsins (*Sebé-Pedrós et al., 2018*; *Steger et al., 2022*). Whether cnidarian clock neurons receive environmental information directly or from dedicated sensory neurons, and whether there is anything analogous to a 'central' clock that coordinates rhythmicity throughout the body via neurotransmitter or hormone signaling, is unknown. Related to this is the question of whether the disruptive effects of SC are due to effects on a single pacemaker clock, or due to the desynchronization of multiple clocks in different tissues. Both scenarios are plausible—dissociation of clock gene expression can occur even within single cells (*Schmal et al., 2019*), and differential synchronization of spatially distinct clocks might contribute to organism-level disruptions. Cell-level studies are needed to localize neurons with rhythmic gene expression or possible roles in thermal and light sensing, and to determine whether *Nematostella* possess multiple spatially distinct clocks.

## Implications for the cnidarian molecular clock

The molecular architecture of circadian clocks is understood in bilaterian model systems, but is largely unknown in Cnidaria. *Nematostella* possesses orthologs of the core circadian transcription factors *Clock* and *Cycle*, but not of the canonical repressor of CLOCK, *Period* (*Reitzel and Tarrant, 2009*; *Reitzel et al., 2013*). This suggests that there are major differences in clock architecture between bilaterians and cnidarians (although some bilaterians also lack *Period*; *Stanton et al., 2022*). We identified binding motifs enriched in the promoters of rhythmic genes, suggesting that bZIP, and possibly SOX, family transcription factors may regulate circadian gene expression in *Nematostella* (*Figure 5— source data 3*). Proline and acidic amino acid-rich bZIP (PAR-bZIP) and SOX-family proteins regulate circadian clock outputs in bilaterians (*Cyran et al., 2003*; *Gachon, 2007*; *Cheng et al., 2019*), and *Nematostella* PAR-bZIP and SOX genes were robustly rhythmic during aligned and SC conditions in our data (*Figure 5—source data 2*). Our results also support existing hypotheses about the roles of cryptochrome proteins in the *Nematostella* clock, and raise new questions about the role of the *Clock* gene in this animal.

Our data support the hypothesis that the cryptochrome protein *Cry2* is involved in the core clock, possibly as a negative regulator of *Clock* (*Reitzel et al., 2013*), while *Cry1a* and *Cry1b* are photosensors. *Cry2* expression was driven by both light and temperature cycles, while rhythmic expression of *Cry1a* and *Cry1b* appears to be driven only by light cycles (*Figure 2c*, *Figure 5d*). Other genes suspected to have central roles in the cnidarian clock, including *Helt-like* (*Helt*), *CIPC*, and two PAR-bZIP transcription factors (*PAR-bZIP-a* and *PAR-BZIP-c*; *Reitzel et al., 2010*; *Oren et al., 2015*; *Tarrant et al., 2019*), were strongly rhythmic in our study. Misalignment between light and temperature cycles did not disrupt oscillations of these putative clock genes, as (except for *Cry1b*) they were all strongly rhythmic during both aligned and SC conditions (*Figure 5d*). *CIPC*, *Cry2*, and *PAR-bZIP-c* advanced in phase by 5–8 hr under SC, while *Clock*, *Helt*, and *PAR-bZIP-a* remained in-phase with the light cycle. Thus, although temperature was a stronger driver of rhythmic gene expression than light overall, light probably controls rhythmic expression of some core clock components.

Surprisingly, we found that *Clock* mRNA did not oscillate during a temperature cycle (*Figure 2c*), and was one of only two dozen genes to directly follow the light cycle. Our data, together with evidence that *Clock* expression is induced by blue light and not green light (*Leach and Reitzel, 2020*), suggests that rhythmic *Clock* expression is entirely blue light-driven and is not required for rhythmic behavior. This would be unusual, as we are not aware of any other system in which *Clock* mRNA oscillates in response to one zeitgeber and not another: *Clock* oscillates during both light and temperature cycles in *Drosophila* (*Darlington et al., 1998*; *Glossop et al., 1999*; *Boothroyd et al., 2007*) and fish (*Lahiri et al., 2005*; *Di Rosa et al., 2015*), and *Cycle/Bmal* does the same in mammals (*Chun et al., 2015*). This may suggest that temperature cycles do not entrain circadian rhythms in *Nematostella* in the same way as light. Strictly speaking, we do not even know whether *Clock* is required for circadian rhythms in *Nematostella*, but it is possible that *Clock* exhibits temperature-driven rhythms in a subset of cells, or that temperature acts on *Clock* activity at a level other than transcription.

## Global patterns of gene expression during SC

Work in corals has shown that diurnal gene expression can differ between different constant temperatures (*Wuitchik et al., 2019*); we show that temperature cycles also interact with light cycles to affect

rhythmic gene expression in cnidarians. To our knowledge, this is the first study of transcriptome-wide gene expression during antiphasic light and temperature cycles in any organism. Because 12 hr SC disrupted rhythmic locomotion, we had expected that *Nematostella* might simply express fewer rhythmic genes in these conditions. Instead, while many genes did lose rhythmicity during SC, several hundred genes in turn became rhythmic, and 7% of the transcriptome remained rhythmic (*Figure 5b*). Thus, a substantial portion of the transcriptome was rhythmic even during conditions that severely disrupted rhythmic behavior. Genes that gained rhythmicity during SC had lower mean expression than Align-specific rhythmic genes, although their amplitude did not differ (*Figure 5—figure supplement 1*). It is unclear whether the cycling of SC-specific genes is unimportant, deleterious, or serves some sort of homeostatic function. Gain or loss of rhythmic RNA abundance does not necessarily imply changes to rhythmic transcription, but could be due to rhythmic variation in splicing, degradation, or any other post-transcriptional process (*Lück et al., 2014*; *Zhang et al., 2014*). Indeed, we observed widespread changes to the expression of genes that regulate RNA metabolism and splicing during SC, possibly affecting the rhythmic abundance of many other transcripts. Post-transcriptional processes, including mRNA processing and translation, have also been implicated in the regulation of rhythmic processes in corals (*Wuitchik et al., 2019*), suggesting a general role for post-transcriptional regulation of rhythmicity in cnidarians.

A large portion of *Nematostella*'s transcriptome seems to be regulated by diel temperature cycles, consistent with observations in other ectotherms such as *Drosophila* (*Boothroyd et al., 2007*). Our data do not allow us to distinguish direct, clock-independent effects of temperature from possible influences of temperature on clock-dependent gene expression, and both processes almost certainly contribute to the changes observed here. Some physiological processes, such as autophagy, may be robustly rhythmic because they are strongly synchronized by ambient temperature. However, the amplitude of temperature-responsive genes was reduced during SC, suggesting that some temperature-responsive genes are also either regulated by light or by the clock itself. Furthermore, temperature-responsive genes were enriched for processes known to be under clock control in other animals, including autophagy, ABC transporter activity, and mRNA metabolism (*Ma et al., 2012*; *Hardin and Panda, 2013*; *Pácha et al., 2021*).

*Nematostella* experienced dramatic perturbations to metabolic gene expression during SC. For instance, genes that mediate ribosome biogenesis, one of the most energy-intensive cellular processes (*Warner, 1999*), lost rhythmicity, as did other genes related to RNA and protein metabolism (*Figure 5e*). Some genes related to glucose metabolism gained rhythmic expression during SC, suggesting that clock dysregulation altered the expression of glucose regulatory genes (*Figure 5e*). Even genes that remained rhythmic and followed the temperature cycle, such as those related to endocytosis, vacuolar transport, and autophagy, changed their temporal expression in relation to time of feeding and thus to the period of peak energy availability. Desynchronization of metabolic processes likely has bioenergetic consequences, as circadian clocks regulate metabolic homeostasis and optimize metabolism for periods of high and low energy demand (*Ma et al., 2012*; *Thurley et al., 2017*; *Hurley et al., 2018*). Disruption of clock function is linked to various metabolic diseases in mammalian models and human health (e.g. *Turek et al., 2005*; *Green et al., 2008*; *Nedeltcheva and Scheer, 2014*; *Roenneberg and Merrow, 2016*), and antiphasic light and temperature cycles can substantially reduce growth in plants—possibly because changes to the phase and amplitude of rhythmic gene expression disrupt the coordination of growth-related processes (*Bours et al., 2013*). Similarly, we suggest that the disruption of coordination between genes regulating metabolism likely has deleterious consequences for *Nematostsella*. Future work should connect these results to other physiological endpoints, such as rhythms in respiration rate (*Maas et al., 2016*).

## Ecological implications

*Nematostella* are small, translucent animals that inhabit salt marshes along the eastern seaboard of North America and, invasively, in Northern Europe; they therefore experience wide fluctuations in salinity, temperature, and water chemistry on tidal, diel, and seasonal scales. The anemones live in shallow water (typically 20–30 cm for our local population; *Tarrant et al., 2019*) and burrow in sediment with their oral end and tentacles exposed. Although *Nematostella* live in a tidally-influenced environment, circadian rhythms in behavior and gene expression are much more prominent than circatidal rhythms, even among animals entrained to natural conditions (*Tarrant et al., 2019*). We do

not know the adaptive significance of *Nematostella*'s circadian behavior, but it may be that nocturnality provides protection from predators or UV radiation.

It may be the case that temperature is not a true zeitgeber in *Nematostella*, since the signal for temperature-entrained free-running rhythms was weak and we did not observe free-running molecular rhythms in core clock genes. However, it has been noted that circadian rhythms of cnidarians are 'weaker' than those of bilaterian model organisms (*Hoadley et al., 2016*) because behavioral free-running rhythms are often noisy and variable among individuals, and most gene expression rhythms appear not to persist during free-running (e.g. *Leach and Reitzel, 2019*). Therefore, the apparent weakness of temperature-entrained free-running rhythms in our study may not be surprising (*Figure 2a*). The observed weakness of circadian rhythms could be biological, implying that cnidarians rely more on direct responses to the environment than on entrained rhythms, or technical in nature (e.g. due to use of a noisy behavioral endpoint, or bulk RNA-seq that masks rhythmic gene expression by averaging). Future work with more sophisticated behavioral endpoints may provide stronger evidence for temperature entrainment of circadian rhythms in this animal. In any case, we demonstrate for the first time that diel temperature cycles influence circadian rhythms in a cnidarian (*Figure 2*), with broad effects on gene expression and circadian behavior. Our data provide strong, but not definitive, evidence that temperature cycles, by themselves, entrain circadian rhythms in *Nematostella*.

We have used the phrase 'sensory conflict' to refer to a situation in which two environmental signals provide different phase information, as informed by experiments with only one signal or the other. This framework emphasizes the two cues as independent sources of information about time of day, which can therefore be placed into an irreconcilable relationship from an information content standpoint. Sensory conflict is perhaps most useful as a framework for experimental design, but is of limited utility when interpreting the full range of possible interactions between zeitgebers. For instance, the choice of a reference time series, in which light and temperature are aligned, is somewhat arbitrary (i.e. is the 2 hr offset between light and temperature 'less aligned?'—after all, light and temperature cycles do not line up exactly in nature). More generally, zeitgebers can combine in an infinite number of ways, especially when one cue (in this case temperature) varies continuously rather than according to an on/off binary. It is thus appropriate to consider our experimental groups simply as different situations in which light and temperature interact, not necessarily as greater or lesser degrees of conflict. In an ecological context, the shapes of natural light and temperature cycles are much more complex and variable than artificial cycles (e.g. *Yoshii et al., 2009*; *Vanin et al., 2012*), and there are many other temporal signals—including food availability, redox state, and tidal and lunar cycles—that all influence organismal behavior and physiology and can interact with one another in complex ways. Although this complexity is challenging to address experimentally, it is important to characterize clock behavior in multi-zeitgeber systems in order to predict how organisms will respond to anthropogenic changes to zeitgeber regimes, for example due to artificial light at night (ALAN). ALAN alters the relationship between light and other signals in the environment and is increasingly affecting ecosystems such as coral reefs (*Gaston et al., 2013*). ALAN impacts the behavior and physiology of many organisms through direct effects of light as well as disruption of circadian rhythms, but we know little about what the consequences might be for any given species. Sensory conflict specifically addresses the effects of phase mismatches between zeitgebers, which is one possible effect of ALAN.

We found that *Nematostella*'s behavior during entraining conditions was 'normal' (i.e. nocturnal) over a wide range of temperature offsets (up to 8 hr; *Figure 3a*). This may indicate that behavioral rhythms are likely to remain in phase with light in most realistic scenarios. On the other hand, behavior was disrupted at the most extreme offsets, indicating that light does not strictly override temperature and there can be interactions between light and temperature cycles that prevent or weaken rhythmic activity. More work could be done to see how light pollution scenarios (e.g. different periods of light, or dim light throughout the night) interact with naturalistic temperature cycles, with the ultimate goal of managing human impacts by implementing minimally disruptive light regimes. Such studies should also consider sublethal effects on gene expression and metabolism that may impact organismal fitness, as we observed substantial changes to gene expression during SC (*Figure 5*). In general, our results illustrate that disruption of circadian rhythms is not an all-or-nothing situation. Some individuals maintained rhythmic behavior even when behavior was severely disrupted on average, and SC

did not abolish rhythmic expression of core clock genes. Instead, SC caused clock outputs to become weaker and less coordinated among individuals. Our findings highlight the need for multi-zeitgeber studies to employ fine temporal resolution across a variety of conditions in order to understand how animal behavior and physiology vary according to the interactions of multiple environmental signals.

# Materials and methods

**Key resources table**

| Reagent type (species) or resource | Designation | Source or reference | Identifiers | Additional information |
|---|---|---|---|---|
| Commercial assay or kit | Aurum Total RNA Fatty and Fibrous Tissue Kit | Bio-Rad | #7326830 | |
| Commercial assay or kit | iScript cDNA synthesis kit | Bio-Rad | #1708891 | |
| Commercial assay or kit | iTaq Universal SYBR Green Supermix | Bio-Rad | #1725120 | |
| Commercial assay or kit | Aurum Total RNA Mini Kit | Bio-Rad | #736820 | |
| Software, algorithm | LinRegPCR | *Ruijter et al., 2009* | | |
| Software, algorithm | NORMA-Gene | *Heckmann et al., 2011* | | |
| Software, algorithm | DeepLabCut | *Mathis et al., 2018* | | |
| Software, algorithm | R | R Project for Statistical Computing | RRID:SCR_001905 | |
| Sequence-based reagent | Clock_F | This paper | PCR primers | TAACCCGGAAGCTGA ATTTG |
| Sequence-based reagent | Clock_R | This paper | PCR primers | GCTTGGGGAAGACAC TAACTTG |
| Sequence-based reagent | Cry2_F | This paper | PCR primers | GCATCTGATTTGCA GAAATGG |
| Sequence-based reagent | Cry2_R | This paper | PCR primers | CTACACGGGCGA GATAGTGG |
| Sequence-based reagent | Cry1a_F | This paper | PCR primers | GCATGAATTCTG GCAGCTGG |
| Sequence-based reagent | Cry1a_R | This paper | PCR primers | CCAACTTCCACA GGGCAGAA |
| Sequence-based reagent | Cry1b_F | This paper | PCR primers | GATTCGGATGTTT GTCGCCA |
| Sequence-based reagent | Cry1b_R | This paper | PCR primers | TCGAACGAGTCCAG TGAACA |
| Sequence-based reagent | Helt_F | This paper | PCR primers | CGGACAAGGGCGCT AATGAA |
| Sequence-based reagent | Helt _R | This paper | PCR primers | CAAGGCTGTTGAGGG TCCAT |

## Animal culture

Animals used in this study were adult anemones of mixed sex. The laboratory population was originally collected from Great Sippewissett Marsh, MA USA and maintained in the laboratory for several generations. *Nematostella* were kept in glass water dishes containing half-strength 1 µm-filtered seawater (from Buzzards Bay, MA), diluted 1:1 with distilled water to a salinity of approximately 16. Water was changed weekly. Prior to acclimating to experimental conditions, anemones were kept at room temperature (18 °C) and fed brine shrimp nauplii four times per week.

## Behavioral experiments

Anemones were acclimated to a given light and temperature regime for at least two weeks prior to behavioral monitoring. They were fed during the acclimation period, but not during behavioral monitoring. A schematic of the experimental design is given in *Figure 1*. For all experiments, animals were randomly selected from the full acclimated group.

In the first set of experiments, *Nematostella* were kept in gradually changing (ramped) 24 hr temperature cycles in constant darkness. Incubators were programmed such that the water temperature reached the desired temperature on the hour, which was monitored with HOBO temperature loggers. Temperature cycles were either 14–26 °C (changing 1 degree per hour) or 8–32 °C (2 degrees per hour). Animals were fed with the aid of a dim red-light headlamp *Nematostella* behavior is most sensitive to blue light

and least sensitive to red light (*Reitzel et al., 2010*; *Leach and Reitzel, 2020*). Animals were transferred to the Noldus chambers at ZT6 (14–26 °C), or ZT8-10 (8–32 °C). Trial lengths ranged from 69 to 73 hr. For the 14–26 °C cycle, the free-running temperature was 20 °C, which was held constant beginning at ZT6. For the 8–32 °C cycle, the free-running temperature was 24 °C, beginning at ZT8. We also conducted light and temperature cycle experiments with short periods (12 hr): a 6:6 light-dark cycle at 20 °C, and a 17–23 °C temperature cycle changing by 1 degree per hour.

For sensory conflict (SC) experiments, animals were acclimated to the 14–26 °C temperature cycle and a 12:12 light-dark cycle. The population of anemones was maintained in an incubator with 'aligned' light and temperature cycles, such that ZT0 corresponded to lights-on and the coldest temperature. In addition to Aligned-cycle, 6 light and temperature regimes were tested in which the phase of the temperature cycle was delayed relative to the light cycle in 2 hr increments, as shown in *Figure 1b*. Anemones in these groups were transferred from the aligned incubator to a second incubator where the temperature cycle was shifted relative to the light cycle. Behavioral monitoring was performed for separate groups of animals during entrainment and free-running conditions, resulting in 14 total groups.

After acclimation, behavior was monitored using Noldus Daniovision observation chambers equipped with infrared-sensitive cameras (DVOC-0040, Noldus Information Technology). Anemones were fed the day before each behavioral trial. A few hours after feeding, 12 anemones were transferred to individual wells of two 6-well plates and left in the incubator overnight. This was done to avoid directly handling the animals at the start of the experiment. At the start of each trial, plates were placed into two Noldus chambers. We chose a minimum sample size of n=24 anemones per group because this is twice the sample size (n=12) at which we regularly detect rhythms in LD; we thus tested four 6-well plates per group (except Off12-cycle, which had a sample size of n=36). A custom flow-through water system was used to control the temperature. A separate tank was heated and cooled using an aquarium heater and cooler connected to a programmable temperature controller, and water was flowed around the plate inside the observation chamber using aquarium pumps. Water temperature was monitored with iButton temperature loggers. Each trial began at ZT6, and was recorded for 78h. Free-running trials begain at ZT12, 6h after the start of recording, and thus had a length of 72h. To avoid suddent jumps in temperature, it was necessary to release groups into constant temperature (20°C) at different times (see *Figure 1b*; the beginning of free-run per se, meaning lights-off and 20°C, was the same for every group). Videos were recorded with a framerate of 2 frames per second.

## qRT-PCR

Anemones were maintained in a 8–32 °C temperature cycle in constant darkness and sampled every 4h over 48h (13 time points), beginning at ZT12. In parallel, another group of animals was moved to constant temperature (24 °C) at ZT8 and sampled at the same time points. Four biological replicates were sampled at each time point, and each replicate consisted of three pooled individuals. Total RNA was extracted using the Aurum Total RNA Fatty and Fibrous Tissue Kit (Bio-Rad) following manufacturer's protocol with DNAse treatment. RNA quality and concentration were checked with a NanoDrop spectrophotometer. Libraries with low 260/230 ratios (<1.8) or low concentration were re-purified using an RNA Clean and Concentrator kit (Zymo). RNA libraries were diluted to $20\,\mathrm{ng\,L^{-1}}$ and reverse transcription was performed with an iScript cDNA synthesis kit (Bio-Rad). Primers for five genes—*Clock*, *Cry1a*, *Cry1b*, *Cry2*, and *Helt*—were ordered from Thermo Fisher and diluted to 10µM. Quantitative PCR (qPCR) was performed with iTaq Universal SYBR Green Supermix (Bio-Rad). The 20µL reaction mixture consisted of 10µL Supermix, 8µL nuclease-free water, 1µL cDNA, and 0.5µL each of forward and reverse primers. All samples for a given primer pair were performed on a single 96-well plate. Annealing temperatures were tested with a gradient for each primer pair followed by a melt curve analysis. 60°C was chosen as the annealing temperature because it resulted in the lowest $C_q$ values with specific amplification. The thermocycler was programmed as follows: 95° for 3 min, 40 cycles of 95° for $10\,\mathrm{s}$ and 60° for $30\,\mathrm{s}$, followed by a melt curve analysis.

Quantitative PCR data analysis was performed with LinRegPCR (*Ruijter et al., 2009*). We excluded a total of three samples from further analysis because several genes failed to amplify. Gene expression was normalized using the NORMA-Gene algorithm, which uses a least-squares approach to minimize variation within treatments rather than relying on reference genes (*Heckmann et al., 2011*). Rhythmicity was tested with Lomb-Scargle periodograms (LSP) implemented in the R package 'lomb' v2.0, with periods between 20 and 28 hr. Significance was assessed with n=2000 permutations, with a slight modification to the default

'randlsp' command. Since each time point contained replicates, all 49 data points (3–4 replicates x 13 time points) were shuffled, and then the mean time point values were re-calculated for the permuted data.

## RNA extraction and sequencing

Anemones were either maintained in Aligned-cycle or Off12-cycle conditions (*Figure 1c*). Animals were sampled every 4 h over 48 h (13 time points), beginning at ZT6, and immediately preserved in RNA-later. Each time point contained three biological replicates of five pooled individuals each, with the exception of one library that consisted of three pooled individuals. This was because the initial five-animal pool produced a low yield, so a new pool was constructed from three remaining animals from that treatment group. RNA was extracted using the Aurum Total RNA mini kit following the manufacturer's instructions, except we incubated the RNA lysis buffer on ice for 30 min following tissue homogenization. Concentrations and quality were assessed with a Nandrop spectrophotometer and a Qubit fluorometer. Libraries were sequenced using 3' Tag-seq to give 100 bp single-end reads at the UT Austin Genomic Sequencing and Analysis Facility (GSAF) on a NovaSeq 6000.

## Data analysis, behavior

Animal coordinates were extracted from raw video files using DeepLabCut v2.1 (DLC, *Mathis et al., 2018*). Each video with footage of six individual anemones was cropped into six videos to be analyzed separately. A training dataset was constructed from 3699 images, in which we manually labeled the center-point of individual anemones. The DLC algorithm was trained on this dataset for 600,000 iterations using default settings and the 'resnet_50' neural network. DLC output files were analyzed using R code derived from the DLCAnalyzer Github package (https://github.com/ETHZ-INS/DLCAnalyzer; *Sturman et al., 2020*; *von Ziegler and Roessler, 2023*); custom code used for this paper is available at this github repository: https://github.com/caberger1/Sensory-Conflict-in-Nematostella-vectensis (copy archived at *Berger, 2023*). Frames with a likelihood less than 0.90 were removed and interpolated, as were frames where the recorded movement was more than 2 cm per frame. An animal was considered 'moving' if its speed was at least $0.03 \, \mathrm{cm \, s^{-1}}$ for a period of at least 3 s (6 frames); any other movement was considered noise and ignored for downstream analysis. For each animal, distance moved was summed into hourly bins based on real-world clock times and normalized to that animal's maximum hourly movement. This corrects for differences in overall activity due to for example body size (as in *Hendricks et al., 2012*; *Oren et al., 2015*; *Tarrant et al., 2019*). For each experimental group, we averaged the hourly time points to produce a mean time series, which we analyzed alongside the individual time series. Time series were smoothed with a centered moving average in 4 h windows.

## Statistics

Two complementary rhythmicity tests were used. The first was a Lomb-Scargle periodogram (LSP) implemented within the R package 'lomb', with periods restricted from 20 to 28 hr and significance assessed by permutation (n=2000). The second test, the empirical JTK method, was implemented within the BioDare2 web portal ('eJTK Classic'; https://biodare2.ed.ac.uk/). This approach fits a series of cosinor waves with 24 hr periods, thus testing for periods of exactly 24 hr. An LSP test with periods from 10 to 14 hr was used to test for circatidal rhythms. In all cases, p-values were adjusted by Benjamin-Hochberg correction, and we chose a conservative FDR cutoff of 0.001 because we were only interested in time series that could confidently be identified as rhythmic. The 'mFourFit' method implemented in the BioDare2 web portal was used to calculate periods and phases of time series. This is a curve-fitting procedure that is generally the most accurate of the available methods for short time series (*Zielinski et al., 2014*).

The R package 'CircaCompare' v0.1.1 was used to test for differences in amplitude and phase of mean time series. CircaCompare uses a cosinor curve-fitting approach and thus gives different estimates of phase and amplitude than those calculated by MFF; however, CircaCompare allows for formal hypothesis testing. The Rayleigh test for circular uniformity (*Rayleigh, 1880*) was used to test whether phases were distributed non-uniformly (implemented in the R package 'circular' v0.4). A bootstrapped version of Watson's non-parametric test (described in *Fisher et al., 1993*; *Pewsey et al., 2013* and implemented in the R package 'AS.circular' v0.0.0.9) was used to test whether phase distributions differed in their mean direction; significance was assessed with 9999 bootstraps. The Kruskal-Wallis test followed by Dunn's test for multiple comparisons were used to test for differences in non-circular means between groups (implemented in R packages 'stats' v4.0.3 and 'dunn.test' v1.3.5). Levene's test was used to test for differences

in variance ('car' v3.0), and paired Wilcoxon rank tests ('stats' v4.0.3) were used to test for differences in distance to expected light and temperature phases. Unless otherwise noted, p-values were corrected for multiple testing using the Benjamini-Hochberg procedure.

Linear mixed effects models were used to test for differences in (un-normalized) activity levels between groups, considering two measures of activity: percent time active, and total distance covered. Each hour of activity for each individual was considered a separate observation. We accounted for baseline differences in activity between individuals using random intercepts, and for autocorrelation of residuals using a correlation structure of AR(1); using more complex correlation structures did not affect the results. Mixed models were implemented in the R package 'nlme' v3.1, and post-hoc testing was done using 'emmeans' v1.6.2.

Time series clustering analysis was done by decomposing smoothed mean time series using wavelet transformation in the R package 'biwavelet' v0.20.21, with the smallest scale (period) set to 2 and the largest scale to 36. The 'wclust' command was used to construct a distance matrix of the 14 wavelet series, and principal component analysis (PCA) was performed on the distance matrix with 'FactoMineR' v2.4. Hierarchical clustering was performed in principal component space with the 'HCPC' command using complete linkage. The AU test was used to quantify uncertainty in clustering in the pvclust v2.2 R package (*Suzuki and Shimodaira, 2006*) with 1000 bootstraps, and clustering and PCA analyses were visualized with the 'factoextra' package v1.0.7.

## Data analysis, gene expression

Raw reads were trimmed using Cutadapt v3.3 (*Martin, 2011*) with a minimum length of 25 and quality cutoff of 5, and mapped to the Simrbase Nvec200_v1 genome (*Zimmermann et al., 2020*) using Salmon v1.5.1 with 'selective alignment (full)' (SAF) (*Srivastava et al., 2020*), a k-mer size of 21, and no length correction (as Tag-seq reads do not have length bias). Tximport v1.16.1 (*Soneson et al., 2016*) was used to import counts into R; counts were then TMM-normalized (*Robinson et al., 2010*) and expressed as counts per million on a log2 scale. Raw reads from *Oren et al., 2015* were re-analyzed in the same way as above, except they were length-corrected.

Differential expression (DE) analysis was performed using limma-voom 3.44.3 (*Phipson et al., 2016*) with quality weights (*Liu et al., 2015*), using the following model: 'Condition * Light +Temperature + hour', allowing for linear effects of temperature and total time in the experiment. Low counts were filtered using the command 'filterByExpr' in EdgeR v3.30.3 grouped by time point and condition, which retained 12,294/15,290 genes. Rhythmic gene expression was tested using RAIN v1.24.0 (*Thaben and Westermark, 2014*), ECHO v4.0.1 (*De Los Santos et al., 2020*), and DryR v1.0.0 (*Weger et al., 2021*). In RAIN, search was performed for waveforms with 24h periods and asymmetries in 4h increments. p-Values were corrected using the Bonferroni procedure. Phases and amplitudes of rhythmic genes were calculated using CircaCompare. Rao's two-sample test for homogeneity (*Jammalamadaka et al., 2021*) was used to test whether groups of phases were drawn from the same distribution (R package 'TwoCircular' v1.0), and significance was assessed with 10,000 Monte-Carlo replications. Discriminant analysis of principal components was conducted with the R package 'adegenet' v2.1.4 and code modified from *Dixon et al., 2015*.

Gene ontology enrichment analysis was performed with the GO_MWU R package with default settings (https://github.com/z0on/GO_MWU, *Wright et al., 2015*; *Matz, 2022*). GO annotations were downloaded from the SimrBase website (https://simrbase.stowers.org/analysis/294, NVEC200.20200813.gff). KEGG enrichment was performed with 'clusterProfiler' v3.0.4. To annotate KEGG pathways, protein sequences from an earlier *Nematostella* genome assembly (Nvec1, JGI) were downloaded from the KEGG website (https://www.genome.jp/kegg/) and queried against the SimrBase transcriptome using TBLASTN. SimrBase genes were annotated with the KEGG term of their best-scoring hit in each pathway, using an e-value cutoff of $1 \times 10^{-10}$ and requiring >75% sequence identity. Sliding window enrichment analysis was performed to identify GO and KEGG terms enriched at certain times of day in each treatment. In each 4 h sliding window (e.g. ZT0-4, ZT3-6), p-values were calculated (using GO_MWU for GO, and clusterProfiler for KEGG) by comparing genes with phase in that window against all other genes. The enrichment score of a gene at each hourly bin was calculated by averaging the -log10 adjusted p-values from the three sliding windows that overlapped the time point (e.g. the score at ZT3 averaged the sliding windows ZT0-4, ZT1-5, and ZT2-6), and a gene was considered enriched if the score at any time point was greater than -log10(0.05).

Weighted Gene Co-expression Network Analysis v1.70 (WGCNA, *Langfelder and Horvath, 2007*) was used to cluster rhythmic genes. For the full network constructed from all genes rhythmic in either time series, a soft-thresholding power of 5 was used, and a power of 8 was used for both the Align- and SC-specific networks. In all cases, a signed hybrid adjacency matrix, a minimum module size of 20, biweight midcorrelation ('bicor'), maxPOutliers = 0.2, and mergeCutHeight = 0.25 were used. Eigengene rhythmicity was assessed using LSPs with 2000 permutations (periods 20–28 hr). Enriched GO terms were calculated for each module using GO_MWU with module membership ('kME') as input and all other genes as background. KEGG enrichment was performed using a hypergeometric test within 'clusterProfiler', as above.

Putative promoter sequences 1 kb upstream of transcription start sites were searched for enriched motifs using the HOMER motif discovery tool (findMotifs.pl), searching for motifs of lengths 6, 8, 10, and 12, retaining the top 15 motifs, and using all other promoters as background. Motifs were considered enriched at a p-value of $1 \times 10^{-5}$ for 'known' motifs, or $1 \times 10^{-10}$ for de novo motifs.

## Acknowledgements

We are particularly grateful to Dr. Carolyn Tepolt for generously providing the aquarium heater, cooler, and temperature controllers, and Dr. Kirstin Meyer-Kaiser for use of her incubator. We also thank Drs. Gregory Fournier, Casey Dunn, Carolyn Tepolt, and Yisrael Schnytzer for helpful discussions.

## Additional information

### Funding

| Funder | Grant reference number | Author |
| --- | --- | --- |
| Woods Hole Oceanographic Institution | Ocean Ventures Fund | Cory A Berger |

The funders had no role in study design, data collection and interpretation, or the decision to submit the work for publication.

### Author contributions

Cory A Berger, Conceptualization, Data curation, Funding acquisition, Investigation, Visualization, Methodology, Writing - original draft; Ann M Tarrant, Resources, Supervision, Writing - review and editing

### Author ORCIDs
Cory A Berger http://orcid.org/0000-0002-6003-1955
Ann M Tarrant http://orcid.org/0000-0002-1909-7899

### Decision letter and Author response
Decision letter https://doi.org/10.7554/eLife.81084.sa1
Author response https://doi.org/10.7554/eLife.81084.sa2

## Additional files

### Supplementary files
• MDAR checklist

### Data availability
Raw RNA-seq data have been uploaded to the NCBI Sequence Read Archive (SRA), Bioproject PRJNA826898. R code used for analysis is available at https://github.com/caberger1/Sensory-Conflict-in-Nematostella-vectensis (copy archived at *Berger, 2023*).

The following dataset was generated:

| Author(s) | Year | Dataset title | Dataset URL | Database and Identifier |
|---|---|---|---|---|
| Berger CA, Tarrant AM | 2022 | Sensory conflict in *Nematostella vectensis* | https://www.ncbi.nlm. nih.gov/bioproject/? term=PRJNA826898 | NCBI BioProject, PRJNA826898 |

The following previously published dataset was used:

| Author(s) | Year | Dataset title | Dataset URL | Database and Identifier |
|---|---|---|---|---|
| Oren M, Tarrant AM, Alon S, Simon-Blecher N, Elbaz I, Appelbaum L, Levy O | 2015 | *Nematostella vectensis* diurnal transcriptomes | https://www.ncbi.nlm. nih.gov/bioproject/? term=PRJNA246707 | NCBI BioProject, PRJNA246707 |

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

## Appendix 1

### Comparison of RAIN, ECHO, and DryR

90% of rhythmic genes identified by RAIN ($p<0.01$) had an adjusted ECHO p-value less than 0.05, and 94% of genes with an ECHO p-value less than 0.01 had a RAIN p-value less than 0.05. Log p-values of ECHO and RAIN were highly correlated ($r=0.85$). DryR does not calculate p-values, but instead uses model selection to cluster genes based on their expression profile. Briefly, 81% of genes assigned rhythmicity in either time series by DryR were rhythmic in that time series based on RAIN ($p<0.01$), and 93% had a RAIN p-value less than 0.05; 73% of rhythmic RAIN genes ($p<0.01$) were assigned to the corresponding cluster(s) in DryR. Thus RAIN and DryR also showed good agreement, although RAIN tended to identify additional rhythmic genes not clustered by DryR. The output of all three programs can be found in *Figure 5—source data 2*.

## Appendix 2

### Comparison with *Oren et al., 2015*

We compared our results to a previous studies of rhythmic gene expression in Nematostella, *Oren et al., 2015* (https://doi.org/10.1038/srep11418), by re-analyzing their raw data using the same software and methods as in the current study, including mapping to the SimrBase genome. *Oren et al., 2015* sampled anemones for RNA-seq every 4 h for 48 h over a LD cycle at a constant temperature 23. This study provides a reference for genes exhibiting rhythmic expression under a light-dark cycle at constant temperature. They used a Fourier analysis to identify 180 genes with rhythmic diel expression, which included putative clock genes *Clock*, *Helt*, *CIPC*, *Cry1a*, and *Cry2* (*Oren et al., 2015*). Using RAIN on their re-analyzed data, we identified 1498 rhythmic genes at $P<0.05$ (560 genes with $P<0.01$); we used a less stringent p-value cutoff here because their sample size was much smaller than the sample size in our current study. Despite the differences in genome assembly and statistical methodology between the original paper and our re-analysis, we recapitulated some of the most prominent rhythmic genes in their dataset, including the principal genes hypothesized to play a role in the cnidarian clock (Clock, Cry1a, Cry2, Helt-like, and CIPC). The phases of clock genes in these two experiments are shown in 3.

This re-analysis of this previously published LD experiment demonstrates that core clock gene expression is comparable between an LD cycle at constant temperature, and our study of an LD cycle with an aligned temperature cycle. The phases of putative clock genes (Clock, Helt, CIPC, Cry1a, and Cry2) were approximately the same in the aligned time series and in Oren et al. (within 4 h), as were the phases of two PAR-bZIP genes. The only exception was Cry1b, which was rhythmic in the current study and not in the re-analyzed dataset ($P=0.2$).

However, the overlap of all rhythmic genes between these studies was small. Only 477/1498 (32%) genes rhythmic in *Oren et al., 2015* ($P<0.05$) had a rhythmic p-value less than 0.05 in the current study (345 had $P<0.01$). Of these 477 shared genes, just 224 (47%) had phases within 4 h of each other, and 108 (23%) shifted by at least 8 h. However, there was no overall difference between the phase distributions (Rao's test, $P=0.5$). Functional enrichment of the *Oren et al., 2015* dataset revealed few similarities with the current study—only a single GO/KEGG term, "Spliceosome", was expressed at the same time (ZT12-18; *Figure 5—source data 2*). It is important to note that *Oren et al., 2015* used animals derived from a different population than the current study (MD versus MA), and differences in other factors, such as diet and time of feeding, could also affect the rhythmic transcriptome. Finally, as noted above, the *Oren et al., 2015* dataset has a much smaller sample size and thus less power to detect true rhythmic transcripts.

