## [Editor Report]

Understanding the integration and contribution of different combinations of environmental cues to the synchronization of the daily oscillator is important, because it provides insight into how organisms might be able to distinguish (and weight) between irregular (or in the tidal zone highly complex) versus regular individual daily changes of light and temperature. The study, which is thoroughly conducted and provides an impressive amount of experimental and analytical work, dissects the effects of sensory conflict on behavior and gene expression rhythms.

---

## [Decision Letter]

**Decision letter after peer review:**

Thank you for submitting your article "Sensory conflict disrupts circadian rhythms in the sea anemone *Nematostella vectensis*" for consideration by *eLife*.

Your article has been reviewed by three peer reviewers, and the evaluation has been overseen by Kristin Tessmar-Raible as the Reviewing Editor and Christian Rutz as the Senior Editor. The reviewers have opted to remain anonymous.

The reviewers have discussed their reviews with one another, and the Reviewing Editor has drafted this decision letter to help you prepare a revised submission.

Essential revisions:

1) Improvement of the clarity of the presentation of the work: There are multiple comments on this in the individual reviews, which the authors need to consider, but especially: Figure 1 is not intuitively understandable; the whole network analysis part/ Figure 6 needs reworking; and the use of the terminology of 'sensory conflict' requires clarification.

2) The role of temperature as a possible entrainment cue is at present unclear. Thus, an experiment where the authors test if temperature alone can function as an entrainment cue for Nematostella needs to be added. Whatever the outcome, it can't make the work less relevant -- but it will have important implications for the interpretation of the results. We suggest performing T-cycle experiments that change the thermoperiod away from 24 hr (e.g., 10 h warm : 10 h cold) and then assess both behavioral and molecular readouts.

3) The authors should also add a section in the discussion (could be in the "Ideas and Speculation" part) where they embed their work in a more general context: (i) ecologically (How do temperature versus light relate in the natural -- typically intertidal -- environment of Nematostella and what do the authors learn from their work how in nature this interplay of light and temperature will likely impact on the Nematostella chronobiology?); and (ii) from a sensory neuroscience perspective (How could the different cues be processed? Nerve net of cnidarians versus the much more centralized nervous systems of, e.g., *Drosophila*).

4) The authors are strongly advised to carefully consider the points raised in the individual reviews, which are appended below.

Note: Please note that *eLife* has recently adopted the STRANGE framework, to help improve reporting standards and reproducibility in animal behaviour research. In your revision, please consider scope for sampling biases and potential limitations to the generalisability of your findings:

https://reviewer.elifesciences.org/author-guide/journal-policies

https://doi.org/10.1038/d41586-020-01751-5

*Reviewer #1 (Recommendations for the authors):*

There are many interesting elements to this manuscript.

The manuscripts explores the impact of interactions between two different entraining factors in circadian behaviours in the sea anemone, Nematostella. The authors view these interactions in terms of a framework called 'sensory conflict', which sets in motion a logical conflict in the way I view the results. I do not see why the 'sensory conflict' framework is required at all, and the fact all data are explored in such a rigid manner leads me to feel that the real interactions that may be important to Nematostella biology could well be lost. Another factor to consider is that there are many environmental cycles in the natural world in addition to the two explored here, including, but not limited to the lunar cycle, tides/currents, salinity, nutrition, UV/DNA damage, oxidation/reduction cycles… do they all cause conflicts? If they did, nothing would happen. Some may synergize or interact in an entire suite of different manners. The interaction would be the interesting element.

I am not sure you are looking at desynchronization at all. If both entraining signals have independent peaks, and you shift their alignments, you would have more peaks, e.g., more than one peak per 24 hours, one for light and one for temperature. Given the noise of the readout, two peaks may just look like noise/desynchronization. In fact, my explanation fits the data in 3A 12 hour offset quite well. You in fact recover a peak rapidly when going into constant darkness, so going back from two nonaligned inputs to one input, dark, which supports locomotory behaviour. I'd encourage you to think of the two entertainers shifting their alignments, and the consequences on your readout.

I'd argue that every statement that assumes the framework of 'sensory conflict' is a real thing in the animal and that the data show how this sensory conflict is impacted is not grammatically correct. I read the results carefully a number of times and thinking about the results simply as phase shifting does not in any way make the data more difficult to understand, and in fact makes it easier. If the phases of two different inputs were exactly aligned, for example temperature and light, they would still interact. If the phase of one were shifted so that the peak were now at a different point in a 24 hour cycle, they would still interact, presumably in a different manner. Is one conflict and the other not? In my mind they are just different examples, not one being a conflict and the other not a conflict.

L24. "Existing studies are limited in the range of tested zeitgeber relationships, and also in their taxonomic breadth, which is restricted to insects and vertebrates among animals. " This is clearly incorrect. There are in fact papers going back over a decade looking into many interacting entraining factors in the same cnidarian phylum, e.g., PMID: 31294494 and quite a volume of papers looking at interactions between two different light entrainers, daylight and moonlight, e.g., PMID: 17947585. There are also some very good papers on tide and light interactions in molluscs (PMCID: PMC3179051), season and temperature and tide (PMID: 33736335), lots on marine isopods, polychaete worms and I'm sure many, many others. PMID: 31294494 even looks at interactions between multiple entrainers, including temperature and light at the transcriptome level. These may not have applied the authors 'sensory conflict' framework, but they look at interactions between inputs, and seem more likely to be biologically relevant in my view. The authors do indeed have some interesting results and have done experiments not covered in the literature just mentioned, but they should look at their new results in light of these previous analyses.

Where the results in Figure 3 replicated? These are the core data that are the foundation of the manuscript. As the results do not align well with the authors' own previous constant temperature experiments, this replication would seem particularly important. I may have simply missed this somehow.

These two statements both occur in the legend to Figure 1: ZT0 refers to the coldest time point AND ZT0 always refers to lights-on. Which is true? Also, in this figure, what does the color coding of animals indicate?

The format of Figure 1B needs to be improved, which I simply do not understand. What was the feeding regimen of the animals, and could it impact the results? What was changed, and when, is not clear from the figure at all. The second part of Figure 1B does not make sense to me; I do not understand the concept of the schematic. Why do the periodograms flatline at midpoint through the light period in freerun? I suggest a major reconfiguration and more detail in the legend.

L183. I do not see the point of the masking effects discussion, though it would be fair to say I rarely do. Same with L434 and elsewhere.

Does the NORMA-Gene algorithm used for qPCR utilize control genes? None are mentioned. Without understanding the method used to control trusting the qPCR cycles isn't possible.

As discussed above, my major issue with the manuscript is the 'sensory conflict' focus that I believe makes the results very difficult to understand. Other reviewers may not share this view, but I think it devalues the excellent and interesting data in an unnecessary way.

*Reviewer #3 (Recommendations for the authors):*

The study represents a large dataset of complex experiments designed to better understand the molecular response to sensory conflict in a basal metazoan. While the study reports a large number of intriguing results, the sheer volume of information makes it difficult to follow the results and discussion. This manuscript would benefit from additional streamlining that could help the flow and readability of the results. For example, the Discussion section of this manuscript is only three pages long, with no references back to individual figures or panels to help readers understand the authors' logic for the conclusions that have been drawn. In contrast, the Results section is roughly seven pages (not including figures) and much of it is extremely difficult to follow. This is particularly challenging given the large number of experiments and parameterizations that are utilized to explain the results. This discrepancy between results and discussion makes me wonder if all the results are indeed discussed, or if some may be better placed in supplemental information, allowing room for more in-depth discussion of the key results.

[Editors' note: further revisions were suggested prior to acceptance, as described below.]

Thank you for resubmitting your work entitled "Sensory conflict disrupts circadian rhythms in the sea anemone *Nematostella vectensis*" for further consideration by *eLife*. Your revised article has been evaluated by Christian Rutz (Senior Editor) and a Reviewing Editor.

The manuscript has been improved significantly, Reviewer 2 however, raised some remaining aspects that we ask you to address, as they are particularly relevant for the chronobiological interpretation of your work.

*Reviewer #2 (Recommendations for the authors):*

The authors did an excellent job addressing my concerns and suggestions and I think the quality of the manuscript has improved considerably. I have only a few minor remarks listed below. My main concern that temperature cycles cannot by themselves entrain the Nematostella circadian clock has been addressed by an additional frequency demultiplication experiment. The results of this experiment are convincing: Animals kept in 6h : 6h temperature cycles display a main rhythmic component with a ~25 h period, showing that the rhythm is not simply driven by temperature. Since the animals also show no free-running circadian rhythms after temperature entrainment, it can also be ruled out that the observed 25 h period simply reflects free-running behaviour one would expect under constant conditions. Nevertheless, I still don't think it is fair to say that temperature cycles entrain the circadian clock of Nematostella. In my view the authors show that temperature cycles clearly influence or modulate the clock, but they do not entrain it. By definition, true entrainment must fulfil two criteria. First, the period of the observed rhythm must equal the period of the entraining cycle (here the temperature cycle) and establish a stable phase relationship to each other. Second, after release into constant conditions, the observed rhythm must continue with the same phase established during the entrainment cycle (e.g., Dunlap et al. 2004; Chronobiology: Biological Timekeeping, Sinauer Associates). None of these criteria are fulfilled (1: no stable phase relationship-the animals fall into two groups, and 2: no rhythm observed in free running conditions after the entrainment cycle), and there are also no molecular rhythms observed in any of the clock genes after temperature cycles. For these reasons I think the authors need to tone down their claim that temperature cycles entrain the Nematostella circadian clock. On the other hand, I think the sensory conflict experiments convincingly show that temperature cycles do influence behaviour and molecular rhythms in combined LD and TC conditions, but again this only demonstrates that temperature cycles influence or modulate the circadian clock. The text should be changed accordingly to reflect these differences.

---

## [Author Response]

Essential revisions:1) Improvement of the clarity of the presentation of the work: There are multiple comments on this in the individual reviews, which the authors need to consider, but especially: Figure 1 is not intuitively understandable; the whole network analysis part/ Figure 6 needs reworking; and the use of the terminology of 'sensory conflict' requires clarification.

We have revised the manuscript to improve clarity, including an overhaul of the conceptual Figure 1, and we have revised the section on network analysis and Figure 6. We also discuss and clarify our use of the phrase “sensory conflict,” and acknowledge that a more holistic conceptual framework is appropriate (L45-61, L613-637).

2) The role of temperature as a possible entrainment cue is at present unclear. Thus, an experiment where the authors test if temperature alone can function as an entrainment cue for Nematostella needs to be added. Whatever the outcome, it can't make the work less relevant -- but it will have important implications for the interpretation of the results. We suggest performing T-cycle experiments that change the thermoperiod away from 24 hr (e.g., 10 h warm : 10 h cold) and then assess both behavioral and molecular readouts.

We added a new experiment to strengthen our claim that temperature cycles, by themselves, entrain circadian rhythms in *Nematostella*. Briefly, we found that animals maintained in a temperature cycle with 12h period (i.e. half the length of a circadian period) had a clear 24h component to their behavior, in addition to (and stronger than) a 12h component. This “frequency demultiplication” is a property of some circadian oscillators entrained to short zeitgeber periods, because the clock essentially perceives every other cycle as a new day. If temperature only drove behavior directly, it would be impossible to observe a 24h behavioral pattern. We also show that this holds for 6:6 light cycles, as expected. We discuss this experiment in more detail in below and in the manuscript.

3) The authors should also add a section in the discussion (could be in the "Ideas and Speculation" part) where they embed their work in a more general context: (i) ecologically (How do temperature versus light relate in the natural -- typically intertidal -- environment of Nematostella and what do the authors learn from their work how in nature this interplay of light and temperature will likely impact on the Nematostella chronobiology?); and (ii) from a sensory neuroscience perspective (How could the different cues be processed? Nerve net of cnidarians versus the much more centralized nervous systems of, e.g., *Drosophila*).

We have lengthened the Discussion section to place our results in the context of *Nematostella*’s ecology (L591-654), and to discuss the processing of sensory cues in the bilaterian nervous systems vs. the non-centralized cnidarian nerve net (L491-504)..

4) The authors are strongly advised to carefully consider the points raised in the individual reviews, which are appended below.

We address the reviewers’ comments point-by-point below.

Reviewer #1 (Recommendations for the authors):There are many interesting elements to this manuscript.The manuscripts explores the impact of interactions between two different entraining factors in circadian behaviours in the sea anemone, Nematostella. The authors view these interactions in terms of a framework called 'sensory conflict', which sets in motion a logical conflict in the way I view the results. I do not see why the 'sensory conflict' framework is required at all, and the fact all data are explored in such a rigid manner leads me to feel that the real interactions that may be important to Nematostella biology could well be lost. Another factor to consider is that there are many environmental cycles in the natural world in addition to the two explored here, including, but not limited to the lunar cycle, tides/currents, salinity, nutrition, UV/DNA damage, oxidation/reduction cycles… do they all cause conflicts? If they did, nothing would happen. Some may synergize or interact in an entire suite of different manners. The interaction would be the interesting element.I am not sure you are looking at desynchronization at all. If both entraining signals have independent peaks, and you shift their alignments, you would have more peaks, e.g., more than one peak per 24 hours, one for light and one for temperature. Given the noise of the readout, two peaks may just look like noise/desynchronization. In fact, my explanation fits the data in 3A 12 hour offset quite well. You in fact recover a peak rapidly when going into constant darkness, so going back from two nonaligned inputs to one input, dark, which supports locomotory behaviour. I'd encourage you to think of the two entertainers shifting their alignments, and the consequences on your readout.I'd argue that every statement that assumes the framework of 'sensory conflict' is a real thing in the animal and that the data show how this sensory conflict is impacted is not grammatically correct. I read the results carefully a number of times and thinking about the results simply as phase shifting does not in any way make the data more difficult to understand, and in fact makes it easier. If the phases of two different inputs were exactly aligned, for example temperature and light, they would still interact. If the phase of one were shifted so that the peak were now at a different point in a 24 hour cycle, they would still interact, presumably in a different manner. Is one conflict and the other not? In my mind they are just different examples, not one being a conflict and the other not a conflict.

Thank you for your thoughtful feedback. Upon reflection, we agree that the phrase “sensory conflict” may be too simplistic, in terms of understanding the many possible relationships between different zeitgebers. We use sensory conflict to refer to a situation in which light and temperature provide different phase information, as informed by experiments with only light and only temperature. This is most useful when thinking about experimental design with two extreme offsets, such as our “aligned” and 12h offset time series—so “conflict” refers to a mismatch in the phase information provided by the individual zeitgebers. We still think this concept is useful because it asks what happens when two signals provide conflicting information about time of day---does the clock prioritize one signal over the other, or do they interact in some more complicated way? However, it’s true that there are many possible relationships between light and temperature that cannot be categorized as “conflict” or “not conflict.”

In any case, we have revised the Introduction and Discussion to discuss these points in some detail, although we have chosen to keep the paper’s title and to use “sensory conflict” when referring to the extreme misalignment of light and temperature. We no longer refer to intermediate offsets as “degrees of sensory conflict.”

When discussing our results, we use the term “desynchronization” to refer to a lack of synchrony between different rhythmic individuals (L192). We are not postulating a mechanism. We simply observe that with large offsets (e.g., in Figure 3A 12h offset), behavior is largely arrhythmic which is caused both by a weakening of the rhythmicity of individual animals as well as a lack of synchronization between the remaining rhythmic individuals (L199-208). Regarding the suggestion that we might expect two behavioral peaks within 24h, we did not observe clear circatidal rhythms in any group, or consistent multiple behavioral peaks in any group or in the behavior profile of any individual. We do in fact discuss the observation that we “recover a peak rapidly when… going back from two nonaligned inputs to one input… which supports locomotory behaviour”—this is the concept of masking that we discuss in L453-467.

L24. "Existing studies are limited in the range of tested zeitgeber relationships, and also in their taxonomic breadth, which is restricted to insects and vertebrates among animals. " This is clearly incorrect. There are in fact papers going back over a decade looking into many interacting entraining factors in the same cnidarian phylum, e.g., PMID: 31294494 and quite a volume of papers looking at interactions between two different light entrainers, daylight and moonlight, e.g., PMID: 17947585. There are also some very good papers on tide and light interactions in molluscs (PMCID: PMC3179051), season and temperature and tide (PMID: 33736335), lots on marine isopods, polychaete worms and I'm sure many, many others. PMID: 31294494 even looks at interactions between multiple entrainers, including temperature and light at the transcriptome level. These may not have applied the authors 'sensory conflict' framework, but they look at interactions between inputs, and seem more likely to be biologically relevant in my view. The authors do indeed have some interesting results and have done experiments not covered in the literature just mentioned, but they should look at their new results in light of these previous analyses.

Thank you for pointing out these references. As you say, our original focus was on papers that explicitly compared 2 or more phase relationships of 2 different zeitgebers with 24h periods—so “sensory conflict” in a somewhat narrow sense. There is certainly a larger literature about interactions between multiple entraining cues in general, and we have revised this section to acknowledge this. That said, in our view many of these studies are not directly relevant to our work; for example, PMID: 31294494 considers the interaction of light cycles with different constant temperatures and does not consider temperature as a zeitgeber. Many of these are field studies and thus do not isolate individual entraining cues.

Where the results in Figure 3 replicated? These are the core data that are the foundation of the manuscript. As the results do not align well with the authors' own previous constant temperature experiments, this replication would seem particularly important. I may have simply missed this somehow.

For each group (subpanel in Figure 3A), a total of n=24 animals were placed in 6-well plates and recorded, with one plate per observation chamber. With 2 observation chambers, we could record 2 plates at a time, so 2 recordings took place over 2 weeks. In this sense, each subpanel consisted of 4 plate-level replicates. For the 12h offset, which we viewed as our most important comparison, we expanded the sample size to 36 just in case the apparent lack of rhythmicity was due to sample size limitations. We have revised the Methods to clarify this.

We also do not think the results do not align with constant temperature experiments. In fact, when light and temperature were closely aligned (e.g. top 3 groups in Figure 3A), behavior had the same phase in activity as in Oren et al. 2015 (PMCID: PMC4476465) and Tarrant et al. 2019 (PMID: 31611292).

These two statements both occur in the legend to Figure 1: ZT0 refers to the coldest time point AND ZT0 always refers to lights-on. Which is true? Also, in this figure, what does the color coding of animals indicate?

Throughout the manuscript, ZT0 always refers to lights-on, EXCEPT in the case of the temperature cycle experiments (in which there is no light cycle). We have clarified this in the text, and we have also revised Figure 1 altogether (the color indicated separate animals used for cycling and free-running experiments).

The format of Figure 1B needs to be improved, which I simply do not understand. What was the feeding regimen of the animals, and could it impact the results? What was changed, and when, is not clear from the figure at all. The second part of Figure 1B does not make sense to me; I do not understand the concept of the schematic. Why do the periodograms flatline at midpoint through the light period in freerun? I suggest a major reconfiguration and more detail in the legend.

We have entirely revised Figure 1. The feeding regime is explained in the Methods, and we have added further explanation there and in the legend (all animals were fed at the same time of day during acclimation). Presumably, feeding at different times would affect the results, essentially creating a 3-zeitgeber experiment.

The red lines represent temperature, not periodograms (hence the constant temperature during free-run). We have clarified this in the legend. As explained in the Methods, it was necessary for each group to reach constant temperature at a different time due to the timing of the temperature cycle relative the light cycle.

L183. I do not see the point of the masking effects discussion, though it would be fair to say I rarely do. Same with L434 and elsewhere.

We believe that masking is an important concept to discuss in our paper, particularly for a general audience that may not be familiar with circadian biology. We would like to convey the fact that behavior is the combined product of direct environmental effects and internal rhythms. Masking (and, conversely, gating) are concepts that help interpret differences in behavior between free-running and cycling conditions, and we think this is useful for understanding Figure 3A, specifically.

Does the NORMA-Gene algorithm used for qPCR utilize control genes? None are mentioned. Without understanding the method used to control trusting the qPCR cycles isn't possible.

NORMA-Gene does not use reference genes, but instead uses a least-squares approach to minimize variation within treatments. We acknowledge that not everyone may be used to this approach, but it is widely applied and has been shown to perform well (e.g. https://doi.org/10.1093/labmed/lmx035). Reference gene methods assume that the reference genes are unaffected by the experiment and measured accurately, but these assumptions can be difficult to meet (https://doi.org/10.1186/1471-2105-10-110, https://doi.org/10.1186/1471-2105-12-250).

It is not our intention to wade into this debate, but we argue that it would be extremely difficult to find reference genes with stable expression across daily cycles (especially temperature cycles), and thus using reference gene-free methods is probably the best approach in this case. We have added a line to the Methods explaining the algorithm (L725-727).

As discussed above, my major issue with the manuscript is the 'sensory conflict' focus that I believe makes the results very difficult to understand. Other reviewers may not share this view, but I think it devalues the excellent and interesting data in an unnecessary way.

As discussed above, we have revised the manuscript to discuss a more holistic framework.

Reviewer #3 (Recommendations for the authors):The study represents a large dataset of complex experiments designed to better understand the molecular response to sensory conflict in a basal metazoan. While the study reports a large number of intriguing results, the sheer volume of information makes it difficult to follow the results and discussion. This manuscript would benefit from additional streamlining that could help the flow and readability of the results. For example, the Discussion section of this manuscript is only three pages long, with no references back to individual figures or panels to help readers understand the authors' logic for the conclusions that have been drawn. In contrast, the Results section is roughly seven pages (not including figures) and much of it is extremely difficult to follow. This is particularly challenging given the large number of experiments and parameterizations that are utilized to explain the results. This discrepancy between results and discussion makes me wonder if all the results are indeed discussed, or if some may be better placed in supplemental information, allowing room for more in-depth discussion of the key results.

Thank you for your comments. We have attempted to improve the flow of the manuscript by expanding this Discussion section (including references to figures), and also by shortening the network analysis section of the Results. We believe that our revised Results section supports our main points in the Discussion, and we hope our revisions make these connections more clear.

[Editors' note: further revisions were suggested prior to acceptance, as described below.]

Reviewer #2 (Recommendations for the authors):The authors did an excellent job addressing my concerns and suggestions and I think the quality of the manuscript has improved considerably. I have only a few minor remarks listed below. My main concern that temperature cycles cannot by themselves entrain the Nematostella circadian clock has been addressed by an additional frequency demultiplication experiment. The results of this experiment are convincing: Animals kept in 6h : 6h temperature cycles display a main rhythmic component with a ~25 h period, showing that the rhythm is not simply driven by temperature. Since the animals also show no free-running circadian rhythms after temperature entrainment, it can also be ruled out that the observed 25 h period simply reflects free-running behaviour one would expect under constant conditions. Nevertheless, I still don't think it is fair to say that temperature cycles entrain the circadian clock of Nematostella. In my view the authors show that temperature cycles clearly influence or modulate the clock, but they do not entrain it. By definition, true entrainment must fulfil two criteria. First, the period of the observed rhythm must equal the period of the entraining cycle (here the temperature cycle) and establish a stable phase relationship to each other. Second, after release into constant conditions, the observed rhythm must continue with the same phase established during the entrainment cycle (e.g., Dunlap et al. 2004; Chronobiology: Biological Timekeeping, Sinauer Associates). None of these criteria are fulfilled (1: no stable phase relationship-the animals fall into two groups, and 2: no rhythm observed in free running conditions after the entrainment cycle), and there are also no molecular rhythms observed in any of the clock genes after temperature cycles. For these reasons I think the authors need to tone down their claim that temperature cycles entrain the Nematostella circadian clock. On the other hand, I think the sensory conflict experiments convincingly show that temperature cycles do influence behaviour and molecular rhythms in combined LD and TC conditions, but again this only demonstrates that temperature cycles influence or modulate the circadian clock. The text should be changed accordingly to reflect these differences.

After considering your points, we agree that we should soften our interpretation and have revised the manuscript to reflect the language that temperature cycles modulate the clock, without necessarily entraining it. In several places, we have changed “entrain” to “influence” or “zeitgebers” to “environmental signals,” etc., and added an explicit discussion of the limitations of our data in terms of temperature entrainment (L603-618).

However, we also believe that our evidence of temperature entrainment is perhaps stronger than you have suggested. Our data do in fact support the first entrainment criteria for 24h temperature cycles (our Figure 2)—anemones have 24h behavioral periods and a stable phase relationship with the temperature cycle. In our interpretation, the fact that phases fall into two groups during the 6:6 temperature (and light) cycles does not invalidate this. Individuals do have a stable phase relationship TO THE CYCLE, it’s just that the cycle itself is a 12h period. This is explained in the text in L142-149.

It's true that our evidence for free-running rhythms is more ambiguous—although, again, we note that several individual animals did have temperature-entrained free-running rhythms, and there were statistically detectable free-running rhythms on average (L120, L128). In our opinion, the main difficulty is the noise of behavioral readouts in this system. We suspect that new behavior assays might provide a stronger signal for the phase of individual animals and free-running rhythms.

For these reasons, in our revised manuscript we acknowledge the limitations of our data but still suggest that our results provide evidence that temperature cycles entrain circadian behavior—even if it is not strictly conclusive.

The major changes are outlined below:

Abstract:

L14-19. Here, we show that temperature cycles modulate circadian locomotor rhythms in *Nematostella vectensis*, a model system for cnidarian circadian biology. We conduct behavioral experiments across a comprehensive range of light and temperature cycles and find that*Nematostella*’s circadian behavior is disrupted by chronic misalignment between light and temperature, which involves disruption of the endogenous clock itself rather than a simple masking effect.

L22-24. Our results show that a cnidarian clock relies on information from light and temperature, rather than prioritizing one cue over the other.

Introduction:

L45-47. We use the phrase "sensory conflict" to refer to a situation in which two environmental signals provide different phase information, as compared to a control in which both signals are aligned in phase.

Results:

L110-111. We first showed that gradual, environmentally-relevant temperature cycles drive rhythmic locomotor behavior and influence circadian rhythms in Nematostella.

L254-256. To gain insight into the relative influence of light and temperature on behavioral rhythms, we compared the phases of rhythmic individuals against the offset between light and temperature (Figure 3d).

Discussion:

L429-441. We show here that ecologically relevant temperature cycles drive rhythmic behavior and influence circadian rhythms in the cnidarian *Nematostella vectensis*, and explore how the relationship between simultaneous light and temperature cycles affects rhythmic behavior and gene expression. Given that white light exerts strong direct effects on behavior in *Nematostella*, including masking free-running locomotor rhythms (Oren et al., 2015; Tarrant et al., 2019), we thought behavior might simply synchronize to the light cycle to the exclusion of temperature. However, *Nematostella*'s behavior became severely disrupted and arrhythmic on average when light and temperature cycles were offset by 10-12h (Figure 3a). Free-running rhythms were also disrupted, depending on the specific phase relationship. This shows that both light and temperature interact to set the phase of *Nematostella*'s clock, and that neither cue dominates the other. As a consequence, normal rhythmic behavior is only possible under certain relationships between environmental signals. The disruptive effects of sensory conflict (SC) were not acute responses to transient conditions, but chronic changes measured after weeks of exposure and acclimation.

L476-478. We also observed prominent non-linear dynamics in free-running behavior, with phase differences of as little as two hours between light and temperature resulting in completely different behavioral outcomes.

L531-543. Surprisingly, we found that Clock mRNA did not oscillate during a temperature cycle (Figure 2c), and was one of only two dozen genes to directly follow the light cycle. Our data, together with evidence that Clock expression is induced by blue light and not green light (Leach and Reitzel, 2020), suggests that rhythmic Clock expression is entirely blue light-driven and is not required for rhythmic behavior. This would be unusual, as we are not aware of any other system in which Clock mRNA oscillates in response to one zeitgeber and not another: Clock oscillates during both light and temperature cycles in *Drosophila* (Darlington et al., 1998; Glossop et al., 1999; Boothroyd et al., 2007) and fish (Lahiri et al., 2005; Di Rosa et al., 2015), and Cycle/Bmal does the same in mammals (Chun et al., 2015). This may suggest that temperature cycles do not entrain circadian rhythms in *Nematostella* in the same way as light. Strictly speaking, we do not even know whether Clock is required for circadian rhythms in Nematostella, but it is possible that Clock exhibits temperature-driven rhythms in a subset of cells, or that temperature acts on Clock activity at a level other than transcription.

L603-618. It may be the case that temperature is not a true zeitgeber in *Nematostella*, since the signal for temperature-entrained free-running rhythms was weak and we did not observe freerunning molecular rhythms in core clock genes. However, it has been noted that circadian rhythms of cnidarians are “weaker” than those of bilaterian model organisms (Hoadley et al., 2016) because behavioral free-running rhythms are often noisy and variable among individuals, and most gene expression rhythms appear not to persist during free-running (e.g. Leach and Reitzel, 2019). Therefore, the apparent weakness of temperature entrained free-running rhythms in our study may not be surprising (Figure 2a). The observed weakness of circadian rhythms could be biological, implying that cnidarians rely more on direct responses to the environment than on entrained rhythms, or technical in nature (e.g. due to use of a noisy behavioral endpoint, or bulk RNA-seq that masks rhythmic gene expression by averaging). Future work with more sophisticated behavioral endpoints may provide stronger evidence for temperature entrainment of circadian rhythms in this animal. In any case, we demonstrate for the first time that diel temperature cycles influence circadian rhythms in a cnidarian (Figure 2), with broad effects on gene expression and circadian behavior. Our data provide strong, but not definitive, evidence that temperature cycles, by themselves, entrain circadian rhythms in *Nematostella*.

L619-631. We have used the phrase “sensory conflict” to refer to a situation in which two environmental signals provide different phase information, as informed by experiments with only one signal or the other. This framework emphasizes the two cues as independent sources of information about time of day, which can therefore be placed into an irreconcilable relationship from an information content standpoint. Sensory conflict is perhaps most useful as a framework for experimental design, but is of limited utility when interpreting the full range of possible interactions between zeitgebers. For instance, the choice of a reference time series, in which light and temperature are aligned, is somewhat arbitrary (i.e. is the 2h offset between light and temperature “less aligned?”—after all, light and temperature cycles do not line up exactly in nature). More generally, zeitgebers can combine in an infinite number of ways, especially when one cue (in this case temperature) varies continuously rather than according to an on/off binary. It is thus appropriate to consider our experimental groups simply as different situations in which light and temperature interact, not necessarily as greater or lesser degrees of conflict.